# BENCHMARKING AND ANALYZING MONOCULAR GEOMETRY ESTIMATION MODELS

## ABSTRACT

Recent advances in discriminative and generative pretraining have yielded geometry estimation foundation models with strong generalization capabilities. While most discriminative monocular geometry estimation methods rely on large-scale fine-tuning data to achieve zero-shot generalization, several generative-based paradigms show the potential of achieving impressive generalization performance on unseen scenes by leveraging pre-trained diffusion models and fine-tuning on even a small scale of synthetic training data. Frustratingly, these models are trained with different recipes on different datasets, making it hard to find out the critical factors that determine the evaluation performance. To resolve the above issue, (1) we build fair and strong baselines in a unified codebase for evaluating and analyzing the state-of-the-art (SOTA) geometry estimation models from pre-training style, fine-tuning data, and model architecture perspectives; (2) we thoroughly evaluate geometry models on challenging benchmarks with diverse scenes and high-quality annotations. Under the fair training and evaluation configuration, our results reveal that stochastic diffusion-based protocol is not optimal for fine-tuning generative-based geometry estimation methods. One-step finetuning and inference protocol is sufficient for generative-based depth and surface normal estimation. Besides, we find that both discriminative and generative pretraining can generalize well under small-scale fine-tuning high-quality data in scale-invariant depth estimation task. DINOv2-pretrained discriminative models achieve slightly higher performance than generative counterparts with *the same small amount of synthetic data.* Furthermore, we have observed that metric depth estimation requires significantly more fine-tuning data than scale-invariant depth estimation for learning the depth scale distribution. We hope this work will inspire future geometry estimation research in building more high-quality fine-tuning datasets and designing more powerful geometry estimation models.

## 1 INTRODUCTION

Monocular depth and surface normal estimation, also referred to as "monocular geometry estimation", poses a fundamental yet intricate challenge of inferring distance and surface orientation from a single image. Its significance is underscored by its broad utility across various downstream tasks, including object detection (Huang et al., 2022; Wang et al., 2020b; Ding et al., 2020), visual navigation (Tateno et al., 2017; Yang et al., 2020; Sun et al., 2022; Yang et al., 2018), novel view synthesis (Deng et al., 2022; Roessle et al., 2022), controllable image generation (Zhang et al., 2023; Esser et al., 2023; Zhao et al., 2024), and 3D scene reconstruction (Sun et al., 2021; Denninger & Triebel, 2020). The importance of this task has led to a significant body of research, resulting in numerous models (Birkl et al., 2023; Yang et al., 2024a; Yin et al., 2023; Hu et al., 2024; Ke et al., 2024) over the past decade.

Although a large number of monocular geometry estimation models exist, they can be divided into two paradigms, *i.e.*, discriminative-based and generative-based. Discriminative monocular geometry estimation models leverage the pre-train priors from fully-supervised image classification backbones, *e.g.*, ConvNeXt (Woo et al., 2023), EfficientNet (Tan & Le, 2019) and ViT (Dosovitskiy et al., 2020), or self-supervised backbones. *e.g.*, DINOv2 (Oquab et al., 2024), previous best discriminative depth estimation models, *i.e.*, DepthAnything (Yang et al., 2024a) and Metric3D (Hu et al., 2024), achieve remarkable generalization performance by fine-tuning DINOv2 backbone with *a large scale of fine-tuning data.* Generative geometry estimation models (Ke et al., 2024; Fu et al., 2024; Xu et al.,

2024; Gui et al., 2024; Lee et al., 2024) unleash the power of pre-trained text-to-image diffusion models, *e.g.*, Stable Diffusion (SD) (Rombach et al., 2022). Several generative geometry estimation models (Ke et al., 2024; Fu et al., 2024; Xu et al., 2024; Gui et al., 2024) show strong generation capability with even *a small-scale high-quality synthetic fine-tuning data*.

However, none of the previous works have systematically investigated the performance of these geometry estimation methods with fair and faithful comparison. The reason is twofold. Firstly, *the different selections of datasets and training configurations hinder the fair evaluations of the newly designed methodologies.* **(1)** The performance distinction for different generative-based finetuning paradigms is unclear. It is hard to evaluate whether the actual improvement is from the algorithmic perspective or the data perspective since they are trained on different datasets and different training configurations. **(2)** The performance distinction between discriminative and generative geometry estimation models when trained on the same scale and quality of data also remains unclear. Secondly, *existing popular geometry estimation benchmarks may not reveal the real performance of the models.* NYUv2 (Silberman et al., 2012) and ScanNet (Dai et al., 2017) are still popular in the evaluation of indoor monocular depth estimation. However, they are collected by an older Kinect-v1 system with noisy depth measurements and noisy imaging for RGB patterns, with only $640 \times 480$ resolution. DIODE (Vasiljevic et al., 2019) and ETH3D (Schops et al., 2017) collect both outdoor and indoor scenes with high-quality data while with low diversity scenes for evaluation. KITTI (Geiger et al., 2012) collects depth maps from the LIDAR sensor and focuses on outdoor driving scenes. For surface normal evaluation, NYUv2 (Silberman et al., 2012), ScanNet (Dai et al., 2017), iBims-1 (Koch et al., 2018), Sintel (Butler et al., 2012) and Virtual KITTI (Gaidon et al., 2016) are widely used by generating surface normal maps from the ground truth depth maps. However, the depth noises in NYUv2 (Silberman et al., 2012), ScanNet (Dai et al., 2017) and iBims-1 (Koch et al., 2018) yield unsatisfactory surface normal ground truth. The limited scene diversity of synthetic datasets, *i.e.*, Sintel (Butler et al., 2012) and Virtual KITTI (Gaidon et al., 2016), cannot evaluate the robustness of the surface normal estimation model for in-the-wild geometry reconstruction. Overall, the existing geometry benchmarks are hindered by two main issues: ground-truth quality and scene diversity. This lack of fair and comprehensive benchmarks can significantly impede the development of geometry estimation research.

To address the aforementioned problems, we perform a comprehensive geometry estimation benchmarking study from two perspectives. **(1) Training strategy.** We reimplement a bunch of SOTA algorithms in a unified codebase, including Marigold (Ke et al., 2024), Geowizard (Fu et al., 2024), GenPercept (Xu et al., 2024), DepthFM (Gui et al., 2024), DMP (Lee et al., 2024), Depth-Anything (Yang et al., 2024a), Depth Anything V2 (Yang et al., 2024b), Metric3D v2 (Hu et al., 2024) and DSINE (Bae & Davison, 2024). As such, we can fairly evaluate their performance under the same training configuration, and figure out whether the performance improvement is coming from the model architecture or coming from the high-quality training data. Previous generative geometry models are all based on Stable Diffusion 2.1 (Rombach et al., 2022) with limited training data, we further explore the potential of generative geometry models by conducting model size scale-up ablations in Table 6. **(2) More benchmark datasets.** Apart from traditional geometry evaluation benchmarks, we build more diverse scenes with high-quality labels for geometry evaluation. For depth estimation, we introduce three extra benchmark datasets, InSpaceType (Wu et al., 2023), MatrixCity (Li et al., 2023), and Infinigen (Raistrick et al., 2023). InSpaceType is an indoor depth evaluation benchmark, which contains 12 scenes, 1260 images, and $2208 \times 1242$ resolution. It is a good complement for indoor benchmarks like NYUv2 and ScanNet. MatrixCity is a rendered dataset with real city-scale scenes, we select 808 street images and 403 aerial images for evaluation. It is suitable for evaluating driving and city scenes. Infinigen is also a high-quality rendered dataset, which contains diverse nature scenes. We use it to verify the generalization capability of depth estimation foundation models in wild scenes. For surface normal estimation, we expand existing benchmark datasets with more high quality and diverse datasets, *e.g.*, indoor MuSHRoom dataset (Ren et al., 2024), outdoor Tank and Temples (T&T) dataset (Knapitsch et al., 2017)[1], and wild Infinigen (Raistrick et al., 2023) dataset.

With the unified codebase, training data, and comprehensive benchmark datasets, we conduct a series of analytical experiments. We surprisingly find that **(1)** The synthetic-to-real domain gap (Maximov et al., 2020) is largely addressed through large-scale discriminative and generative pretraining. In other words, it is now feasible to use only synthetic fine-tuning data to achieve generalizable

---

[1]The surface normal annotation of MushRoom and T&T are obtained from Gaustudio (Ye et al., 2024)

performance across diverse real-world scenes. **(2)** It is not necessary for generative-based geometry estimation models, *e.g.*, Marigold (Ke et al., 2024), to follow the original stochastic diffusion protocol due to its inference inefficiency. A simple deterministic one-step fine-tuning protocol is enough to achieve comparable performance. **(3)** For scale-invariant depth estimation, discriminative model with DIONv2 pretraining, and generative model with Stable Diffusion pretraining, are both capable of achieving generalizable performance even with a small-scale fine-tuning dataset. However, the discriminative-based model consistently outperforms the generative-based model across all evaluation benchmarks. **(4)** For metric depth estimation, the benchmark result shows that even initializing the vision encoder with DINOv2 pre-training, it is still impractical to learn generaliable metric depth by fine-tuning only on small-scale datasets. It is consistent with the currently best metric depth estimation model, *i.e.*, Metric3Dv2 (Hu et al., 2024), which focuses on collecting more diverse training datasets (16M training samples) to achieve depth-scale generalization capability. **(5)** For surface normal estimation, both discriminative model DSINE (Bae & Davison, 2024) and generative-based one-step GenPercept achieve impressive results on diverse benchmarks, which suggests appropriate image-level supervision, *i.e.*, inductive bias (Bae & Davison, 2024) for DSINE, and angular loss for GenPercept, is an important factor in providing strong supervision for surface normal estimation task. We hope our benchmarking results could pave the way for designing more powerful geometry estimation algorithms and developing high-quality geometry estimation training datasets in the future.

## 2 PRELIMINARIES

**Task definition.** Given an input image $x \in \mathbb{R}^{H \times W \times 3}$, the goal of monocular geometry estimation is to predict the depth map $d \in \mathbb{R}^{H \times W}$, which can be affine-invariant or metric depth, and surface orientation, which can be represented as either a unit vector $\mathbf{n} \in S^2$, or a 3D *axis-angle* $\mathbf{R} \in SO(3)$.

**Discriminative geometry estimation models.**

With the widespread application of deep learning (LeCun et al., 2015), learning-based methods have demonstrated their ability to estimate geometric information from monocular images (Eigen et al., 2014; Godard et al., 2019; Ranftl et al., 2022; Yang et al., 2024a). Early works primarily relied on discriminative models using either supervised or unsupervised methods. Eigen et al. (Eigen et al., 2014) proposed the first learning-based method for monocular depth estimation, employing two deep network stacks and using ground truth depth for supervision. Zhou et al. proposed an early unsupervised framework, SfMLearner (Zhou et al., 2017), in which camera pose and monocular depth are learned together. With the availability of large amounts of data, recent methods (Ranftl et al., 2022; Yang et al., 2024a; Yin et al., 2023; Hu et al., 2024) have shown a trend toward using large-scale datasets to develop robust geometry estimation models that generalize well to diverse environments. For instance, Ranftl et al. (Ranftl et al., 2022) introduced a method that demonstrates strong zero-shot testing ability by utilizing mixed training datasets. Yang et al. (Yang et al., 2024a;b) further improved zero-shot monocular depth estimation performance by proposing Depth-Anything and Depth-Anything v2, which leverages large-scale pseudo data to achieve strong generalization ability. Meanwhile, Yin et al. (2023); Hu et al. (2024) proposed Metric3D series, which can output accurate metric depth by training models on large-scale public RGB-D datasets and synthetic datasets. Apart from depth estimation, advancements in surface normal information have also been achieved through the use of discriminative models. Surface normal information can not only be calculated directly from depth maps but can also be independently obtained through surface normal estimation techniques (Wang et al., 2015; Ladický et al., 2014; Lenssen et al., 2020; Bae & Davison, 2024). For example, Bae & Davison (2024) proposed a method that demonstrates strong generalization capabilities and produces high-quality surface normal predictions by investigating inductive biases. Overall, the use of discriminative models for both depth and surface normal estimation has shown its significance in improving performance, thereby broadening the applications of monocular geometry estimation.

**Generative geometry estimation Models.** Given the impressive results of recent generative models (Rombach et al., 2022) in image generation tasks, many studies have endeavored to incorporate generative-based pipelines into geometry estimation. Ji et al. (2023) proposed a method to extend the denoising diffusion process into the modern perception pipeline, which can be generalized to most dense prediction tasks, such as depth estimation. Saxena et al. (2024) formulated optical flow and monocular depth estimation as image-to-image translation using generative diffusion models,

Table 1: Quantitative comparison on 5 zero-shot affine-invariant depth benchmarks **with author released weights**. We mark the best discriminative and generative results in bold and the second best underlined. Discriminative methods are colored in blue while generative ones in green .

| Method | Train Samples | Year | NYUv2 | | KITTI | | ETH3D | | ScanNet | | DIODE | |
|---|---|---|---|---|---|---|---|---|---|---|---|---|
| | | | AbsRel ↓ | δ1 ↑ | AbsRel ↓ | δ1 ↑ | AbsRel ↓ | δ1 ↑ | AbsRel ↓ | δ1 ↑ | AbsRel ↓ | δ1 ↑ |
| Metric3D v2 (Hu et al., 2024) | 16M | arXiv'24 | **3.9** | 97.9 | **5.2** | **97.9** | **4.0** | **98.3** | **2.3** | **98.9** | 14.7 | **89.2** |
| DepthAnything (Yang et al., 2024a) | 63.5M | CVPR'24 | 4.3 | **98.0** | 8.0 | 94.6 | 5.8 | 98.4 | 4.3 | 98.1 | 26.1 | 75.9 |
| DepthAnything v2 (Yang et al., 2024b) | 62.6M | arXiv'24 | 4.3 | 97.9 | 8.0 | 94.3 | 6.6 | 98.3 | 4.2 | 97.9 | 32.1 | 75.8 |
| Marigold (Ke et al., 2024) | 74K | CVPR'24 | 5.5 | 96.4 | **9.9** | **91.6** | **6.5** | **96.0** | 6.4 | 95.1 | **30.8** | **77.3** |
| GeoWizard (Fu et al., 2024) | 280K | arXiv'24 | 5.9 | 95.9 | 12.9 | 85.1 | 7.7 | 94.0 | 6.6 | 95.3 | 32.8 | 75.3 |
| GenPercept (Xu et al., 2024) | 74K | arXiv'24 | **5.2** | **96.6** | 10.1 | 90.1 | 6.6 | 95.7 | **5.7** | **96.3** | 31.1 | 76.3 |
| DepthFM (Gui et al., 2024) | 63K | arXiv'24 | 8.2 | 93.2 | 17.4 | 71.8 | 10.1 | 90.2 | 9.5 | 90.3 | 33.4 | 72.9 |

without specialized loss functions and model architectures. Zhao et al. (2023) proposed VPD, a framework that exploits the semantic information of a pre-trained text-to-image diffusion model in visual perception tasks. Ke et al. (2024) introduced a method for affine-invariant monocular depth estimation, where the depth information is derived from retained rich stable diffusion priors. Fu et al. (2024) proposed a foundation model for jointly estimating depth and surface normal from monocular images, which not only achieves surprisingly robust generalization on various types of real or synthetic images but also faithfully captures intricate geometric details. In summary, recent generative-based methods have provided new solutions and demonstrated their applications for depth estimation.

**Geometric evalutaion metrics.** We use widely adopted evaluation metrics for assessing the performance of depth and surface normal estimation. Specifically, for the depth estimation task, we use mean absolute relative error (AbsRel) and accuracy under thresholds ($\delta_i < 1.25^i, i = 1, 2, 3$) for accuracy comparisons. These evaluation metrics for depth estimation are calculated as follows: **(1)** mean absolute relative error (AbsRel): $\frac{1}{n}\sum_{i=1}^{n}\frac{|z_i - z_i^*|}{z_i^*}$; **(2)** the accuracy under threshold ($\delta_i < 1.25^i, i = 1, 2, 3$): % of $z_i$ s.t. $\max\left(\frac{z_i}{z_i^*}, \frac{z_i^*}{z_i}\right) < 1.25^i$; where $z_i$ is the ground truth depth and $z_i^*$ represents the predicted depth. For surface normal estimation, we calculate the angular error for the pixels with ground truth and report both the median and mean values (lower is better). In addition, we measure the percentage of pixels with an error below $t \in [5.0°, 11.25°, 30.0°]$ (higher is better). Please refer to (Bae & Davison, 2024) for calculation details.

## 3 BENCHMARKING DEPTH ESTIMATION FOUNDATION MODELS

### 3.1 A BRIEF OVERVIEW OF SOTA METHODS

To demonstrate the performance of the SOTA methods, we consider some latest and representative algorithms, *i.e.*, two discriminative models, (Metric3Dv2 (Hu et al., 2024), Depth-Anything (Yang et al., 2024a)), and four generative models (Marigold (Ke et al., 2024), DepthFM (Gui et al., 2024), Geowizard (Fu et al., 2024) and GenPercept (Xu et al., 2024)). We fairly evaluate their performance by using the official released model weights on 5 popular benchmarks, *i.e.*, NYU v2 (Silberman et al., 2012), KITTI (Geiger et al., 2012), ETH3D (Schops et al., 2017), ScanNet (Dai et al., 2017) and DIODE (Vasiljevic et al., 2019), in Table 1. Notably, all the methods do not use these benchmarks as training data. We can easily observe that **(1)** Metric3Dv2 (Hu et al., 2024) achieves the best performance on all evaluation datasets, another discriminative-based method, Depth-Anything (Yang et al., 2024a) achieves the second best performance. Both of them are trained on large-scale datasets, with 16M and 63.5M training data separately. **(2)** Generative methods can achieve impressive results on these evaluation benchmarks with even a small amount of fine-tuning data.

In addition to quantitative results, we further test their generalization capability by qualitative visualization in several challenging scenes. Fig. 1 demonstrates the results of three algorithms on line drawing images (left), color draft images (middle), and photo-realistic images (right). Surprisingly, Metric3D fails on both line draw images and color draft images, while Marigold (Ke et al., 2024) and Depth-Anything (Yang et al., 2024a) show some generalization capability on this kind of non-geometrically consistent hand-drawn images. We conjecture that discriminative-based Metric3D

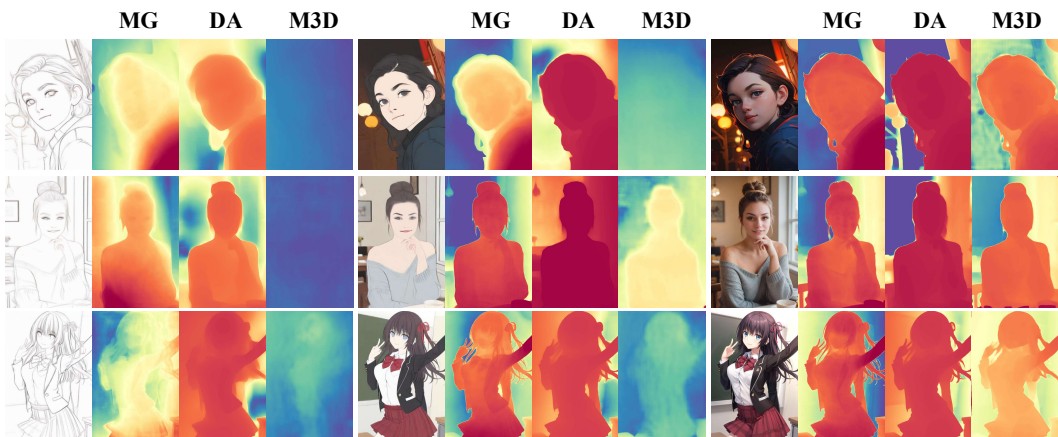

Figure 1: Depth visualization on cartoon images. 'MG' indicates Marigold (Ke et al., 2024), 'DA' indicates Depth-Anything (Yang et al., 2024a), 'M3D' indicates Metric3Dv2 (Hu et al., 2024).

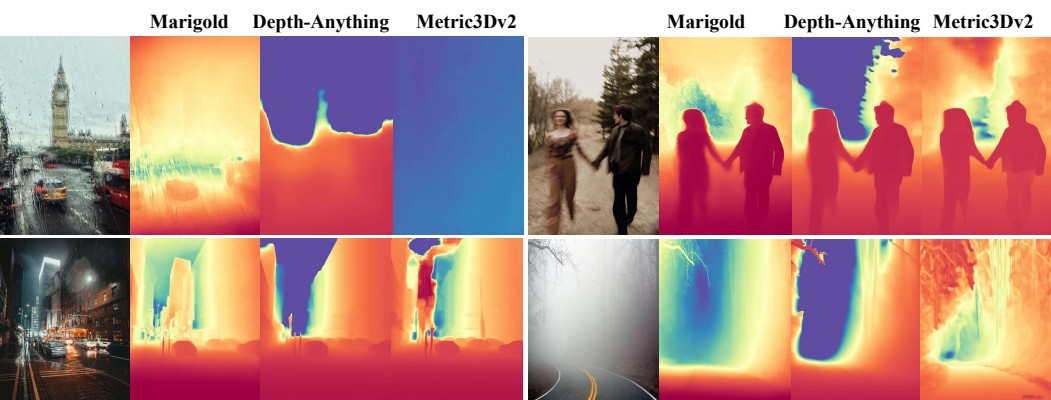

Figure 2: Depth visualization on four challenging scenes, *i.e.*, rainy (top-left), blurry (top-right), dark (bottom-left), and foggy (bottom-right) environments.

does not see cartoon images in the training stage, which leads to poor performance in this scenario. Contrarily, although Marigold (Ke et al., 2024) also does not see cartoon images in their training set, it leverages the priors stored in the pre-trained Stable Diffusion (Rombach et al., 2022) model. Stable Diffusion (Rombach et al., 2022) model has seen millions of text-cartoon pairs when performing text-to-image generation training. Fig. 2 shows the robustness of existing depth estimation models on challenging scenes like rainy, blurry, dark, and foggy environments. Both Metric3D and Depth-Anything fail on the rainy scene; both Marigold and Metric3D fail to estimate the sky in the second blurry scene. None of the algorithms can handle all environments perfectly. Fig. 3 illustrates the depth estimation results on the Infinigen (Raistrick et al., 2023) dataset (first two lines) and BEDLAM (Black et al., 2023) dataset (last line). Infinigen (Raistrick et al., 2023) is a photo-realistic rendered dataset with diverse nature scenes. BEDLAM (Black et al., 2023) is a human-centered high-quality rendered dataset with versatile indoor and outdoor scenes. Mainstream depth evaluation metrics overlook the depth accuracy on the edges of the objects. We use these two datasets to demonstrate the fine-grained depth estimation results since both datasets have high-quality annotations. For measuring the accuracy of depth estimation on edges. We use Canny (Canny, 1986) edge detector to extract the edge mask from the image and then calculate the traditional depth metrics. As shown in Table 2, Depth-Anything achieves the highest performance on the Infinigen dataset; Marigold achieves the best AbsRel on the BEDLAM (Black et al., 2023) dataset.

In a nutshell, discriminative models trained on large data, *i.e.*, Depth-Anything (Yang et al., 2024a), get the highest performance in most cases, while generative models finetuned on small data, *e.g.*, Marigold (Ke et al., 2024), show competitive generalization capability on unseen scenes.

Table 2: Benchmark depth estimation on Infinigen (Raistrick et al., 2023) and BEDLAM (Black et al., 2023) dataset. 'Standard' indicates using standard evaluation metrics. 'Canny' indicates only evaluating the performance on pixels that belong to canny edges. We mark the best results in bold.

| Method | Train Samples | Infinigen-Standard AbsRel ↓ | δ1 ↑ | Infinigen-Canny AbsRel ↓ | δ1 ↑ | BEDLAM-Standard AbsRel ↓ | δ1 ↑ | BEDLAM-Canny AbsRel ↓ | δ1 ↑ |
|---|---|---|---|---|---|---|---|---|---|
| Marigold (Ke et al., 2024) | 74K | 32.9 | 80.9 | 28.0 | 78.7 | **16.2** | 82.4 | **19.6** | 80.3 |
| Metric3Dv2 (Hu et al., 2024) | 16M | 14.5 | 80.7 | 18.6 | 77.8 | 28.1 | **84.7** | 26.3 | **80.8** |
| Depth-Anything (Yang et al., 2024a) | 63.5M | **12.0** | **88.4** | **14.3** | **84.7** | 46.2 | 69.0 | 46.8 | 67.8 |

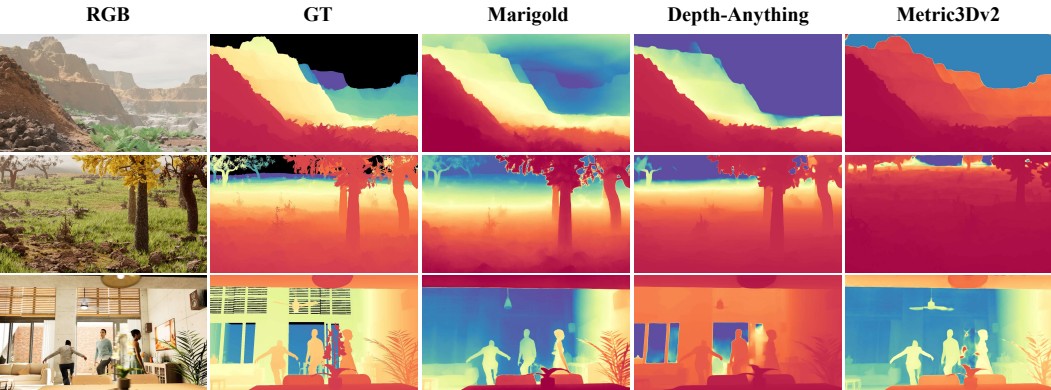

Figure 3: Fine-grained depth estimation comparison. We select two scenes (first two rows) from the Infinigen Dataset (Raistrick et al., 2023) and one scene (last row) from the BEDLAM dataset (Black et al., 2023).

## 3.2 BENCHMARKING DIFFERENT GENERATIVE FINE-TUNING PARADIGMS

Several fine-tuning paradigms have been proposed for diffusion-based depth estimation. Based on network architecture, they can be divided into two categories. The first category methods (Marigold (Ke et al., 2024) and DepthFM (Gui et al., 2024)) concatenate the image latent and depth latent encoded by VAE (Kingma & Welling, 2014) encoder as the input of the UNet latent denoiser. As such, the input channels of the latent denoiser are doubled (8 input channels) to fit the expanded input. The second category methods (DMP (Lee et al., 2024) and GenPercept (Xu et al., 2024)) drop the depth latent, so they follow the original latent denoiser's architecture (4 input channels). Based on fine-tuning paradigms, they can be divided into four categories. **(1)** Marigold (Ke et al., 2024) treats the initial depth latent as standard Gaussian noise and progressively denoise it with the same scheduler as the original Stable Diffusion pipeline. **(2)** DepthFM also treats the initial depth latent as standard Gaussian noise, however, the difference is that they finetune the denoiser with Flow Matching (Lipman et al., 2022) pipeline, with auxiliary surface normal loss. **(3)** DMP (Lee et al., 2024) reformulates the task as a blending process, *i.e.*, translating the image latent to depth latent with the Stable Diffusion v-prediction (Rombach et al., 2022) learning target. **(4)** GenPercept (Xu et al., 2024) further improve the efficiency of DMP (Lee et al., 2024) by proposing a one-step inference pipeline. Based on the amount of fine-tuned parameters, they can be divided into two categories. The first category methods (Marigold, DepthFM, GenPercept) directly fine-tune the UNet parameters. The second category method (DMP) adds LORA (Hu et al., 2021) layers into the UNet architecture to achieve the goal of depth estimation.

In this section, we fairly benchmark the four fine-tuning protocols by training on the Hypersim dataset (38,387 samples), with $480 \times 640$ resolution, $3 \times 10^{-5}$ learning rate, 96 batch sizes, and 10, 000 iterations. We choose Stable Diffusion 2.1 (Rombach et al., 2022) as the base model. As shown in Table 3, **(1)** Fine-tuning all UNet parameters outperforms using LORA layers. (compare line 1 and line 2 on DMP) **(2)** Stochastic Marigold and deterministic GenPercept achieve comparable performance, and outperforms other protocols. This implies that GenPercept's one-step finetuning approach is sufficient for depth estimation. Diffusion-based and flow-based finetuning protocols are not necessary for generative-based geometry estimation models to achieve generalizable performance.

Table 3: Benchmarking different generative finetuing paradigms on 5 zero-shot affine-invariant depth benchmarks. We mark the best results in bold and the second best underlined.

| Method | Train Samples | FT Strategies | NYUv2 AbsRel ↓ | NYUv2 δ1 ↑ | KITTI AbsRel ↓ | KITTI δ1 ↑ | ETH3D AbsRel ↓ | ETH3D δ1 ↑ | ScanNet AbsRel ↓ | ScanNet δ1 ↑ | DIODE AbsRel ↓ | DIODE δ1 ↑ |
|---|---|---|---|---|---|---|---|---|---|---|---|---|
| DMP (Lee et al., 2024) | 38K | LORA | 13.2 | 85.1 | 19.2 | 74.3 | 16.2 | 83.7 | 14.1 | 84.6 | 45.6 | 62.1 |
| DMP (Lee et al., 2024) | 38K | UNet | 10.1 | 90.6 | 15.4 | 80.0 | 10.0 | 91.0 | 10.9 | 89.0 | 38.2 | 68.7 |
| Marigold (Ke et al., 2024) | 38K | UNet | 6.9 | 95.0 | 13.8 | 80.7 | 7.5 | 93.7 | 7.1 | 94.3 | 28.7 | 74.6 |
| GenPercept (Ke et al., 2024) | 38K | UNet | 5.3 | 96.7 | 13.9 | 81.5 | 6.2 | 96.0 | 5.8 | 96.3 | 32.0 | 74.9 |
| DepthFM (Ke et al., 2024) | 38K | UNet | 10.9 | 89.5 | 19.2 | 68.8 | 12.9 | 86.3 | 11.4 | 87.7 | 33.6 | 72.4 |

Table 4: Inference latency and speed benchmark for different components and methods. 'Infer Steps' indicates the minimum repeat times of the U-Net for achieving optimal results. All models are inference with $512 \times 512$ resolution, except CLIP (Radford et al., 2021) ($224 \times 224$).

| Method | Components | Params/M | Macs/GFLOPs | Latency/s | Memory/G | Inference Steps |
|---|---|---|---|---|---|---|
| Depth-Anything (Yang et al., 2024a) | ViT-L (Dosovitskiy et al., 2020) + Head | 335.3 | 586.0 | 0.19 | 2.24 | 1 |
| Metric3Dv2 (Hu et al., 2024) | ViT-L (Dosovitskiy et al., 2020) + Head | 411.9 | 1014.0 | 0.60 | 2.67 | 1 |
| DSINE (Bae & Davison, 2024) | EfficientNet B5 (Tan & Le, 2019) + Head | 72.6 | 38.7 | 0.06 | 0.73 | 1 |
| - | VAE-Tiny (madebyollin., 2023) | 2.4 | 131.9 | 0.03 | 0.61 | 1 |
| - | VAE | 83.7 | 1781.2 | 0.11 | 0.65 | 1 |
| Geowizard (Fu et al., 2024) | CLIP | 304.0 | 77.8 | 0.04 | 1.25 | 1 |
| Marigold-LCM (Ke et al., 2024) | VAE+ UNet | 949.6 | 3138.4 | 0.29 | 5.27 | 4 |
| Geowizard (Fu et al., 2024) | VAE+ UNet + CLIP | 861.2 | 9846.1 | 0.85 | 5.24 | 10 |
| DepthFM (Gui et al., 2024) | VAE+ UNet | 949.6 | 2459.8 | 0.21 | 5.40 | 2 |
| GenPercept (Xu et al., 2024) | VAE+ UNet | 949.6 | 2120.4 | 0.18 | 5.40 | 1 |
| GenPercept (Xu et al., 2024) | VAE-Tiny (madebyollin., 2023) + UNet | 868.3 | 471.1 | 0.18 | 5.40 | 1 |

Table 5: Benchmarking the inference efficiency of Marigold. We mark the best results in bold.

| Method | VAE Version | Infer Steps | NYUv2 AbsRel ↓ | NYUv2 δ1 ↑ | KITTI AbsRel ↓ | KITTI δ1 ↑ | ETH3D AbsRel ↓ | ETH3D δ1 ↑ | ScanNet AbsRel ↓ | ScanNet δ1 ↑ | DIODE AbsRel ↓ | DIODE δ1 ↑ |
|---|---|---|---|---|---|---|---|---|---|---|---|---|
| Marigold (Ke et al., 2024) | base | 50 | 5.5 | 96.4 | 9.9 | 91.6 | 6.5 | 96.0 | 6.4 | 95.1 | 30.8 | 77.3 |
| Marigold-LCM (Ke et al., 2024) | base | 4 | 6.1 | 95.8 | 10.1 | 90.6 | 6.3 | 96.0 | 6.9 | 94.7 | 30.9 | 77.3 |
| Marigold-LCM (Ke et al., 2024) | tiny (madebyollin., 2023) | 4 | 6.9 | 95.0 | 13.8 | 80.7 | 7.5 | 93.7 | 7.1 | 94.3 | 32.8 | 73.8 |
| Marigold-LCM (Ke et al., 2024) | tiny (madebyollin., 2023) | 1 | 6.6 | 95.4 | 13.0 | 83.6 | 7.8 | 93.2 | 7.0 | 94.5 | 33.3 | 73.1 |

Table 6: Benchmarking discriminative and generative depth model with the same training data (77K).

| Network | Pretrain Style | Backbone | NYUv2 AbsRel ↓ | NYUv2 δ1 ↑ | KITTI AbsRel ↓ | KITTI δ1 ↑ | ETH3D AbsRel ↓ | ETH3D δ1 ↑ | ScanNet AbsRel ↓ | ScanNet δ1 ↑ | DIODE AbsRel ↓ | DIODE δ1 ↑ |
|---|---|---|---|---|---|---|---|---|---|---|---|---|
| ViT+DPT Head | Random init | ViT-L | 21.1 | 62.5 | 27.2 | 53.1 | 23.4 | 61.1 | 19.2 | 67.4 | 32.4 | 57.7 |
| ViT+DPT Head | DINOv2 (Oquab et al., 2024) | ViT-L | 4.9 | 97.5 | 8.5 | 94.1 | 8.1 | 97.0 | 5.1 | 97.6 | 24.5 | 74.6 |
| Marigold (Ke et al., 2024) | SD21 (Rombach et al., 2022) | UNet | 6.9 | 95.8 | 12.2 | 85.7 | 9.2 | 95.5 | 7.1 | 95.4 | 25.2 | 73.0 |
| Marigold (Ke et al., 2024) | SDXL (Podell et al., 2023) | UNet | 6.8 | 95.8 | 11.1 | 89.2 | 8.9 | 96.7 | 6.3 | 96.2 | 24.5 | 73.6 |

## 3.3 INFERENCE EFFICIENCY OF DEPTH ESTIMATION FOUNDATION MODELS

Compared to discriminative models, the inference efficiency may become a bottleneck of the generative-based methods. In this section, we give detailed inference efficiency evaluation in Table 4. We can see that discriminative methods have fewer parameters than generative models. The main inference consumption of the generative models happens on VAE (Kingma & Welling, 2014) and multiple inference steps of UNet. The last line of Table 4 shows that GenPercept (Xu et al., 2024) can achieve comparable inference latency with Depth-Anything (ViT-Large) and a tiny VAE encoder (madebyollin., 2023). In Table 5, we found LCM (Luo et al., 2023) can effectively reduce the inference steps of Marigold (Ke et al., 2024) while maintaining the performance. Besides, a pre-trained tiny VAE (madebyollin., 2023) can substitute the standard VAE (Rombach et al., 2022) with a minimal performance loss.

## 3.4 DISCRIMINATIVE AND GENERATIVE DEPTH ESTIMATORS IN THE SAME DATA REGIME

*Can discriminative depth estimation models achieve competitive results with small-scale high-quality training datasets like generative-based methods?* To answer this question, we benchmark discriminative and generative geometry model with the same amount of training data and the same training strategy. Specifically, we use three training datasets, *i.e.*, Hypersim (38,387) (Roberts et al., 2021), Virtual Kitti (16,790) (Gaidon et al., 2016) and Tartanair (31,008) (Wang et al., 2020a), with

Table 7: Benchmarking depth estimation foundation models on more diverse benchmarks. We mark the best results in bold.

| Network | Pretrain Style | Backbone | Train Samples | InspaceType | | MatrixCity | | Infinigen | |
|---|---|---|---|---|---|---|---|---|---|
| | | | | AbsRel ↓ | δ1 ↑ | AbsRel ↓ | δ1 ↑ | AbsRel ↓ | δ1 ↑ |
| Metric3D v2 (Hu et al., 2024) | DINOv2 (Oquab et al., 2024) | ViT-L | 16M | 10.1 | 89.7 | **9.5** | 89.3 | 14.5 | 80.7 |
| Depth-Anything (Yang et al., 2024a) | DINOv2 (Oquab et al., 2024) | ViT-L | 63.5M | **8.2** | 92.9 | 16.4 | **89.7** | 12.0 | 88.4 |
| ViT+DPT Head | DINOv2 (Oquab et al., 2024) | ViT-L | 77K | 8.4 | **94.0** | 28.0 | 82.4 | **11.4** | **89.5** |
| Marigold (Ke et al., 2024) | SD21 (Rombach et al., 2022) | UNet | 77K | 9.2 | 92.7 | 17.0 | 82.9 | 14.1 | 83.9 |

Table 8: Data scale ablation of metric depth estimation. Offical model is colored in blue, while ablation models are colored in blue.

| Method | Train Samples | Dataset | NYUv2 | | KITTI | | ETH3D | | ScanNet | | DIODE | |
|---|---|---|---|---|---|---|---|---|---|---|---|---|
| | | | AbsRel ↓ | δ1 ↑ | AbsRel ↓ | δ1 ↑ | AbsRel ↓ | δ1 ↑ | AbsRel ↓ | δ1 ↑ | AbsRel ↓ | δ1 ↑ |
| Metric3D v2 (Hu et al., 2024) | 16M | mixed | 8.7 | 94.2 | 7.1 | 93.7 | 32.5 | 17.6 | 11.1 | 90.3 | 24.3 | 81.9 |
| Metric3D v2 (Hu et al., 2024) | 39K | hypersim | 10.1 | 92.1 | 12.2 | 90.7 | 36.5 | 31.7 | 14.3 | 83.0 | 48.6 | 18.8 |
| Metric3D v2 (Hu et al., 2024) | 31K | tartanair | 34.9 | 28.4 | 38.7 | 24.0 | 30.0 | 31.3 | 32.8 | 56.8 | 74.6 | 69.3 |
| Metric3D v2 (Hu et al., 2024) | 70K | hypersim+tartanair | 19.2 | 76.0 | 35.1 | 29.1 | 29.4 | 25.6 | 19.3 | 77.5 | 51.9 | 74.1 |
| Metric3D v2 (Hu et al., 2024) | 350K | hypersim+tartanair+paralleldomain4d | 12.1 | 89.4 | 10.2 | 91.3 | 27.6 | 37.2 | 15.6 | 80.9 | 34.1 | 79.8 |

total 77,897 samples. Both models are trained with 20,000 iterations, with a total batch size of 96 on 4 GPUs. For the discriminative depth model, we follow the network architecture of Depth-Anything (Yang et al., 2024a) (ViT-Large backbone pre-trained with DINOv2 and DPT (Ranftl et al., 2021) head), supervised with the affine-invariant loss (Yang et al., 2024a). For the generative geometry model, we choose Marigold (Ke et al., 2024) as our baseline. We can see from Table 6 that (**1**) the discriminative model is largely inferior to generative-based Marigold on all evaluation datasets without DINOv2 pre-train (line 1 *v.s.*line 3). However, the discriminative model beats Marigold by a large margin when initialized with DINOv2 pre-train weight (line 2 *v.s.*line 3); (**2**) scale-up Marigold from SD21 to SDXL brings consistent improvement in all benchmarks. We can see from Table 7 that our discriminative model trained on 77K data outperforms Metric3Dv2 (Hu et al., 2024) in all three datasets, and, is comparable with Depth-Anything (Yang et al., 2024a) in two datasets (InspaceType and Infinigen). This phenomenon suggests that **high-quality fine-tuning data**, rather than large-scale training data or pre-train paradigm, is indispensable for scale-invariant depth estimation models to achieve strong generalizable performance.

### 3.5 DATA-SCALE ABLATION ON METRIC DEPTH ESTIMATION TASK

Given the success of using small-scale synthetic data for scale-invariant depth estimation, we aim to investigate if the same conclusion holds for metric depth estimation. Thus, we perform data-scale ablation studies upon the SOTA metric depth estimation model, Metric3Dv2 (Hu et al., 2024). Specifically, we adopt Metric3Dv2 model with ViT-small backbone, initialized with the Metric3Dv2 pre-train. We supervise the model with all of the loss functions mentioned in the paper (Hu et al., 2024) for 30K iterations. We use three synthetic datasets, Hypersim, Tartanair, and a large-scale synthetic driving scene dataset, ParallelDomain4D (280K) (par, 2024) for data ablation. As shown in Table 8, the performance of Metric3Dv2 model keeps improving as the data scale grows. However, the performance of small-scale dataset finetuning is largely behind the official model trained on 16M data samples. Hence, large-scale datasets with diverse scales and cameras is still indispensable for metric depth estimation.

## 4 BENCHMARKING SURFACE NORMAL ESTIMATION FOUNDATION MODELS

### 4.1 A BRIEF OVERVIEW OF SOTA METHODS

DSINE (Bae & Davison, 2024) and Metric3Dv2 (Hu et al., 2024) are two representative discriminative surface estimation models, which leverage the geometry priors from two distinct perspectives. DSINE leverages two forms of inductive bias: (**1**) per-pixel ray direction, and (**2**) the relationship between the neighboring surface normal, to learn a generalizable surface normal estimator. Metric3Dv2 (Hu et al., 2024) proposes to optimize the surface normal map by distilling diverse data knowledge from the estimated metric depth. Different from discriminative models, GeoWizard (Fu et al., 2024) is a

Table 9: Quantitative evaluation of the generalization capabilities possessed by **different methods with official released weights.** For each metric, the best results are bolded. Discriminative methods are colored in blue while generative ones in green.

| Method | NYU v2 | | | | | | | ScanNet | | | | | | | Sintel | | | | | | |
|---|---|---|---|---|---|---|---|---|---|---|---|---|---|---|---|---|---|---|---|---|---|
| | mean | med | 5.0° | 7.5° | 11.25° | 22.5° | 30° | mean | med | 5.0° | 7.5° | 11.25° | 22.5° | 30° | mean | med | 5.0° | 7.5° | 11.25° | 22.5° | 30° |
| Metric3D v2 (Hu et al., 2024) | **13.5** | **6.7** | **40.1** | **53.5** | **65.9** | **82.6** | **87.7** | **11.8** | **5.5** | **46.6** | **60.7** | **71.6** | **85.4** | **89.7** | **22.8** | **14.2** | **18.4** | **28.5** | **41.6** | **66.7** | **75.8** |
| DINSE (Bae & Davison, 2024) | 16.4 | 8.4 | 32.8 | 46.3 | 59.6 | 77.7 | 83.5 | 18.3 | 9.3 | 27.1 | 42.0 | 56.3 | 75.0 | 81.2 | 32.0 | 23.9 | 9.0 | 15.0 | 23.8 | 47.5 | 59.4 |
| Geowizard (Fu et al., 2024) | 19.8 | 11.2 | 18.0 | 32.7 | 50.2 | 73.0 | 79.9 | 21.1 | 11.9 | 15.9 | 29.7 | 47.4 | 70.7 | 77.8 | 36.1 | 28.4 | 4.1 | 8.6 | 16.9 | 39.8 | 52.5 |

| Method | MuSHRoom Subset (Indoor) | | | | | | | T&T Subset (Outdoor) | | | | | | | Infinigen Subset (Wild) | | | | | | |
|---|---|---|---|---|---|---|---|---|---|---|---|---|---|---|---|---|---|---|---|---|---|
| | mean | med | 5.0° | 7.5° | 11.25° | 22.5° | 30° | mean | med | 5.0° | 7.5° | 11.25° | 22.5° | 30° | mean | med | 5.0° | 7.5° | 11.25° | 22.5° | 30° |
| Metric3D v2 (Hu et al., 2024) | **14.3** | **7.9** | **31.9** | **48.1** | **61.8** | **81.7** | **87.2** | 22.3 | 14.1 | 19.2 | 31.4 | 43.0 | 64.8 | 73.5 | **32.6** | **27.3** | **5.1** | **10.1** | **17.8** | **41.3** | **54.4** |
| DINSE (Bae & Davison, 2024) | 14.8 | 8.6 | 28.1 | 44.6 | 59.7 | 80.4 | 87.0 | **17.3** | **11.0** | **24.2** | **37.3** | **50.6** | **74.1** | **82.4** | 35.9 | 32.6 | 2.1 | 4.6 | 9.8 | 30.5 | 45.1 |
| Geowizard (Fu et al., 2024) | 16.5 | 10.7 | 14.7 | 30.5 | 52.5 | 79.6 | 86.2 | 20.8 | 13.4 | 10.7 | 23.4 | 42.2 | 70.3 | 78.5 | 36.2 | 32.0 | 1.8 | 4.0 | 8.86 | 30.8 | 46.2 |

generative surface normal estimator without using any inductive bias from the geometry priors. It purely relies on pre-trained diffusion priors to estimate the surface normal map. Table 10 summarizes their performance on six benchmarks. The Mushroom (Ren et al., 2024) (indoor), T&T (Knapitsch et al., 2017) (outdoor), and Infinigen (Raistrick et al., 2023) (wild) datasets are constructed by us to add more diverse scenes with accurate surface normal labels in the evaluation benchmarks. We can see that Metric3Dv2 (Hu et al., 2024) outperform DSINE (Bae & Davison, 2024) and GeoWizard (Fu et al., 2024) in most datasets. Note it is an unfair comparison since **(1)** Metric3Dv2 (Hu et al., 2024) is trained on 16M images, while DSINE is trained on 160K images, and GeoWizard is trained on 280K images. **(2)** DSINE use a much smaller backbone, EfficientNet-B5 (Tan & Le, 2019), while Metric3Dv2 (Hu et al., 2024) employs the ViT-Large (Dosovitskiy et al., 2020) backbone.

## 4.2 DISCRIMINATIVE AND GENERATIVE MODELS IN THE SAME DATA REGIME

In this section, we fairly benchmark discriminative DSINE (Bae & Davison, 2024) and several representative generative geometry models, *i.e.*, Marigold (Ke et al., 2024), DMP (Lee et al., 2024), GenPercept (Xu et al., 2024), and DepthFM (Gui et al., 2024), with 5 training datasets, Hypersim (Roberts et al., 2021) $(38, 387)$, Tartanair (Wang et al., 2020a) $(31, 008)$, Virtual Kitti (Gaidon et al., 2016) $(16, 790)$, BlendedMVS (Yao et al., 2020) $(17, 819)$, ClearGrasp (Sajjan et al., 2020) $(22, 720)$, a total of $126, 724$ samples. For generatative-based models, we represent the output surface normals as unit vectors. We follow DSINE (Bae & Davison, 2024) to represent the outputs of discriminative-based model as axis-angles with three degrees of freedom. All models are trained with $20, 000$ iterations, 96 batch sizes, $480 \times 640$ resolution on 4 A800 GPUs. All generative-based models use $3 \times 10^{-5}$ learning rate. For discriminative model, we follow DSINE (Bae & Davison, 2024) to use $3 \times 10^{-5}$ learning rate for the backbone and $3 \times 10^{-4}$ learning rate for the decoder. We can see from Table 9 that **(1)** DSINE can scale up the performance by using ViT-Large backbone with DINOv2 pretrain (compared with ImageNet pretrained Efficient-B5 backbone). **(2)** For generative-based fine-tuning protocols, DepthFM (Gui et al., 2024) outperforms other paradigms in most benchmarks. *We attribute this to the decoder supervision during the training.* Paradigms that requires multi-step denoising inference steps, *e.g.*, Marigold (Ke et al., 2024) and DMP (Lee et al., 2024) are not suitable to perform decoder supervision during the training. To verify the conjecture, we add decoder loss supervision to one-step GenPercept, termed GenPercept*. The results on NYUv2, ScanNet and Sintel datasets show that decoder surface normal loss supervision can largely improve original GenPercept without decoder supervision. **(3)** Discriminative models, equipped with inductive bias, also achieve impressive results. It is promising to inject inductive bias into the diffusion-based models, as such, the surface normal estimator can effectively leverage the diffusion priors and inductive bias to boost the performance. **(4)** DSINE (ViT-Large in Table 9) trained with 120K samples achieves comparable performance with Metric3Dv2 trained with 16M samples (Table 10). The results verify the point that data-quality is more important than the data-scale in surface normal estimation task.

## 5 BENCHMARKING CROSS-VIEW GEOMETRIC CORRESPONDENCE

Can current monocular geometry estimation foundation models improve the 3D awareness of the original representation models, *e.g.*, DINOv2 and Stable Diffusion? To answer the

Table 10: Quantitative evaluation of the generalization capabilities **with the same training data** on different benchmarks. The best results are bolded. GenPercept* indicates with image-level loss supervision after VAE decoder. 'EB5' indicates ImageNet (Deng et al., 2009) pre-trained EfficientNet-B5 (Tan & Le, 2019). 'ViT-L' indicates DINOv2 (Oquab et al., 2024) pre-trained ViT-Large. The best results are **bolded**. The best results of generativate-based models are underlined.

| Method | Backbone | NYUv2 | | | | | | | ScanNet | | | | | | | Sintel | | | | | | |
|---|---|---|---|---|---|---|---|---|---|---|---|---|---|---|---|---|---|---|---|---|---|---|
| | | mean | med | 5.0° | 7.5° | 11.25° | 22.5° | 30° | mean | med | 5.0° | 7.5° | 11.25° | 22.5° | 30° | mean | med | 5.0° | 7.5° | 11.25° | 22.5° | 30° |
| DSINE (Bae & Davison, 2024) | EB5 (Tan & Le, 2019) | 19.2 | 10.0 | 27.1 | 40.1 | 53.9 | 73.8 | 80.1 | 17.3 | 11.0 | 24.2 | 37.3 | 50.6 | 74.1 | 82.4 | 35.9 | 32.6 | 2.1 | 4.6 | 9.8 | 30.5 | 45.1 |
| DSINE (Bae & Davison, 2024) | ViT-L (Oquab et al., 2024) | 16.2 | 8.2 | 32.8 | 46.9 | 60.6 | 78.5 | 84.1 | 16.1 | 7.4 | 34.5 | 50.6 | 63.8 | 79.4 | 84.3 | 24.6 | 16.1 | 11.1 | 21.1 | 35.1 | 63.8 | 74.1 |
| GenPercept (Xu et al., 2024) | UNet (Ronneberger et al., 2015) | 17.4 | 9.5 | 24.1 | 40.5 | 55.9 | 75.7 | 82.2 | 18.5 | 9.4 | 23.0 | 40.2 | 56.7 | 75.4 | 81.3 | 38.6 | 27.1 | 4.5 | 9.1 | 18.0 | 42.5 | 54.2 |
| GenPercept* (Xu et al., 2024) | UNet (Ronneberger et al., 2015) | 16.4 | 8.0 | 33.3 | 47.8 | 60.9 | 78.3 | 83.7 | 15.2 | 7.4 | 33.9 | 50.7 | 65.0 | 80.9 | 85.7 | 34.6 | 26.2 | 5.2 | 9.8 | 18.4 | 43.8 | 55.8 |
| Marigold (Ke et al., 2024) | UNet (Ronneberger et al., 2015) | 20.2 | 10.9 | 21.8 | 36.0 | 51.2 | 72.8 | 79.4 | 20.5 | 10.3 | 19.7 | 36.0 | 53.5 | 73.6 | 79.2 | 41.3 | 28.7 | 5.5 | 11.1 | 19.7 | 40.9 | 51.7 |
| DMP (Lee et al., 2024) | UNet (Ronneberger et al., 2015) | 21.9 | 11.3 | 19.7 | 34.2 | 49.7 | 71.1 | 77.6 | 22.5 | 11.2 | 17.6 | 32.5 | 50.3 | 71.2 | 76.9 | 45.0 | 39.3 | 4.2 | 7.9 | 13.8 | 29.9 | 39.0 |
| DepthFM (Gui et al., 2024) | UNet (Ronneberger et al., 2015) | 17.8 | 9.3 | 27.7 | 41.9 | 56.5 | 76.7 | 82.5 | 18.5 | 8.6 | 28.4 | 44.7 | 58.8 | 75.6 | 81.0 | 34.1 | 25.8 | 7.1 | 13.5 | 22.0 | 44.8 | 55.7 |

| Method | Backbone | MuSHRoom Subset (Indoor) | | | | | | | T&T Subset (Outdoor) | | | | | | | Infinigen Subset (Wild) | | | | | | |
|---|---|---|---|---|---|---|---|---|---|---|---|---|---|---|---|---|---|---|---|---|---|---|
| | | mean | med | 5.0° | 7.5° | 11.25° | 22.5° | 30° | mean | med | 5.0° | 7.5° | 11.25° | 22.5° | 30° | mean | med | 5.0° | 7.5° | 11.25° | 22.5° | 30° |
| DSINE (Bae & Davison, 2024) | EB5 (Tan & Le, 2019) | 17.9 | 10.1 | 23.9 | 38.9 | 53.8 | 75.8 | 82.5 | 21.7 | 15.4 | 13.4 | 25.7 | 39.0 | 64.4 | 75.2 | 36.5 | 32.7 | 2.0 | 4.4 | 9.3 | 29.8 | 44.8 |
| DSINE (Bae & Davison, 2024) | ViT-L (Oquab et al., 2024) | 12.8 | 6.9 | 34.6 | 53.5 | 67.9 | 84.9 | 89.6 | 18.8 | 11.2 | 22.5 | 36.3 | 49.9 | 71.1 | 79.5 | 33.6 | 28.7 | 2.7 | 5.9 | 13.1 | 38.3 | 52.2 |
| GenPercept (Xu et al., 2024) | UNet (Ronneberger et al., 2015) | 15.0 | 7.9 | 32.4 | 48.0 | 62.7 | 80.6 | 86.3 | 27.4 | 14.0 | 17.8 | 30.1 | 43.1 | 63.7 | 71.0 | 38.8 | 33.5 | 2.1 | 4.9 | 10.5 | 31.2 | 44.3 |
| GenPercept* (Xu et al., 2024) | UNet (Ronneberger et al., 2015) | 13.8 | 8.6 | 33.6 | 48.9 | 63.5 | 84.7 | 89.8 | 19.3 | 13.7 | 13.4 | 24.2 | 40.8 | 72.4 | 81.3 | 34.2 | 29.6 | 2.2 | 5.1 | 12.5 | 39.2 | 58.3 |
| Marigold (Ke et al., 2024) | UNet (Ronneberger et al., 2015) | 17.7 | 9.9 | 19.6 | 36.8 | 53.7 | 77.0 | 83.1 | 29.1 | 14.6 | 14.4 | 26.0 | 40.3 | 63.0 | 70.2 | 39.2 | 34.0 | 2.4 | 5.4 | 11.5 | 31.4 | 43.9 |
| DMP (Lee et al., 2024) | UNet (Ronneberger et al., 2015) | 20.4 | 10.0 | 19.3 | 36.0 | 55.2 | 73.8 | 79.1 | 27.7 | 17.9 | 9.1 | 17.2 | 31.9 | 57.9 | 66.1 | 43.1 | 38.1 | 1.7 | 4.0 | 9.5 | 26.3 | 38.1 |
| DepthFM (Gui et al., 2024) | UNet (Ronneberger et al., 2015) | 17.0 | 9.0 | 25.9 | 42.5 | 58.9 | 77.4 | 82.4 | 22.1 | 13.9 | 14.2 | 26.7 | 42.1 | 65.4 | 74.1 | 31.9 | 27.9 | 2.5 | 5.4 | 11.6 | 38.3 | 54.0 |

question, we follow Probe3D (Banani et al., 2024) by using geometric correspondence estimation, since 3D awareness implies consistency of representations across different views. Specifically, given two views of the same scene, geometric correspondence estimation needs to identify pixels across views that depict the same point in 3D space. We extract feature maps from either trained monocular geometry models or representation models, *e.g.*, DINOv2, and directly compute correspondence between the dense feature maps of different views. We use Paired ScanNet (Dai et al., 2017) for scene evaluation and NAVI wild set (Jampani et al., 2024) for object evaluation. Following Banani et al. (2024), we report the correspondence recall, *i.e.*, the

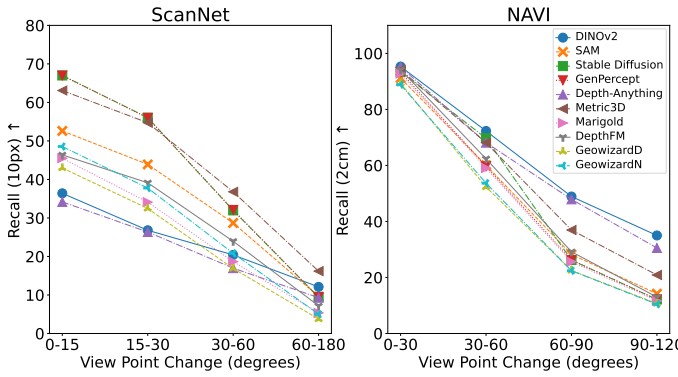

Figure 4: Geometry correspondences evaluation. 'GeowizardD', 'GeowizardN' indicate depth and normal features from Geowizard.

percentage of correspondence that falls within some defined distance. We can see from that 4 **(1)** the discriminative depth estimation model (Depth-Anything with ViT-Large backbone fine-tuned on 77K training samples) fine-tuned from DINOv2 is comparable to the original DINOv2, while generative-based models, *i.e.*, Marigold, DepthFM, GenPercept, and Geowizard, get lower performance than original Stable Diffusion model. **(2)** All models struggle with larger view changes, while generative-based models see a larger drop. In general, monocular geometry estimation models are not 3D-consistent with large viewpoints and thus not yet good enough to encode the 3D structure of the real-world scenario. In other words, it is still an unsolved but promising area to design generalizable pre-training methods that can improve the geometry estimation model's multi-view consistency performance.

## 6 CONCLUSION AND DISCUSSION

In this work, we present the *first* large-scale benchmarking of discriminative and generative geometry estimation foundation models with diverse evaluation datasets. We identify that a strong pre-train model, either Stable Diffusion or DINOv2, combined with high-quality fine-tuning data, is the key to achieving generalizable monocular geometry estimation. Besides, we analyze the critical components for generative-based fine-tuning, and the impacts of datasets' scale and quality in monocular geometry estimation tasks. We believe this benchmarking study can provide strong baselines for unbiased comparisons in geometry estimation studies. Limitations, extra visualizations, and future works are discussed in the Supp. Mat.

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
