# A SUPPLEMENTARY

## A.1 DEPTH ESTIMATION VISUALIZATION OF DIFFERENT METHODS

We visualize the depth estimation results in Fig. 1 and Fig. 2.

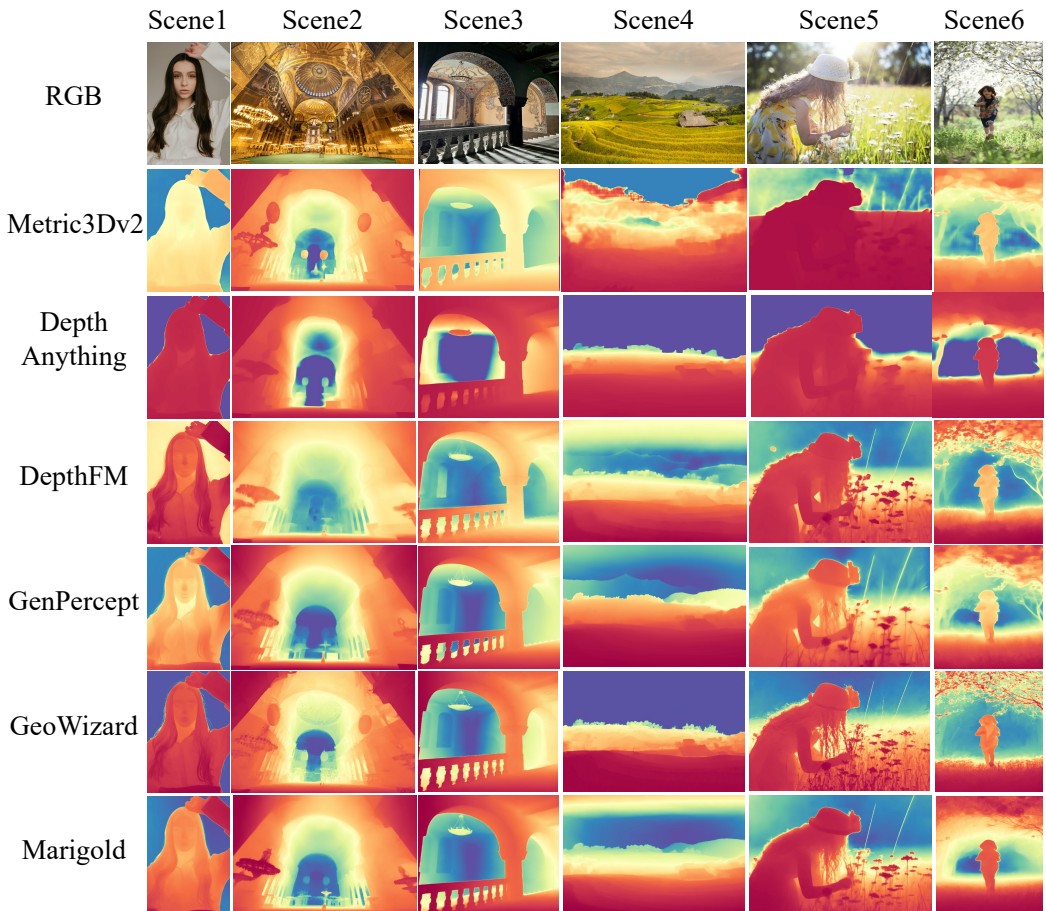

Figure 1: Visualization of different depth estimation methods.

## A.2 SURFACE NORMAL ESTIMATION VISUALIZATION OF DIFFERENT METHODS

We visualize the surface normal estimation results in Fig. 3 and Fig. 4.

## A.3 CORRESPONDENCE ESTIMATION RESULTS

We give more detailed correspondence estimation results in Table 1 for reference. Note that we find that multi-step inference, *e.g.*, 10 steps, can improve the performance of Stable Diffusion in correspondence estimation tasks. Metric3Dv2 Hu et al. (2024) employs DINOv2 with registers Darcet et al. (2023) as the backbone, which has higher performance than DINOv2 without registers Oquab et al. (2024).

## A.4 SURFACE NORMAL ESTIMATION DATASETS

NYUv2 Silberman et al. (2012) is an real indoor dataset comprised RGB-D video sequences from a variety of indoor scenes captured from the Microsoft Kinect. We evaluate on the official test (654 images) set with the ground-truth surface normal generated by Ladicky *et al*. Ladickỳ et al. (2014).

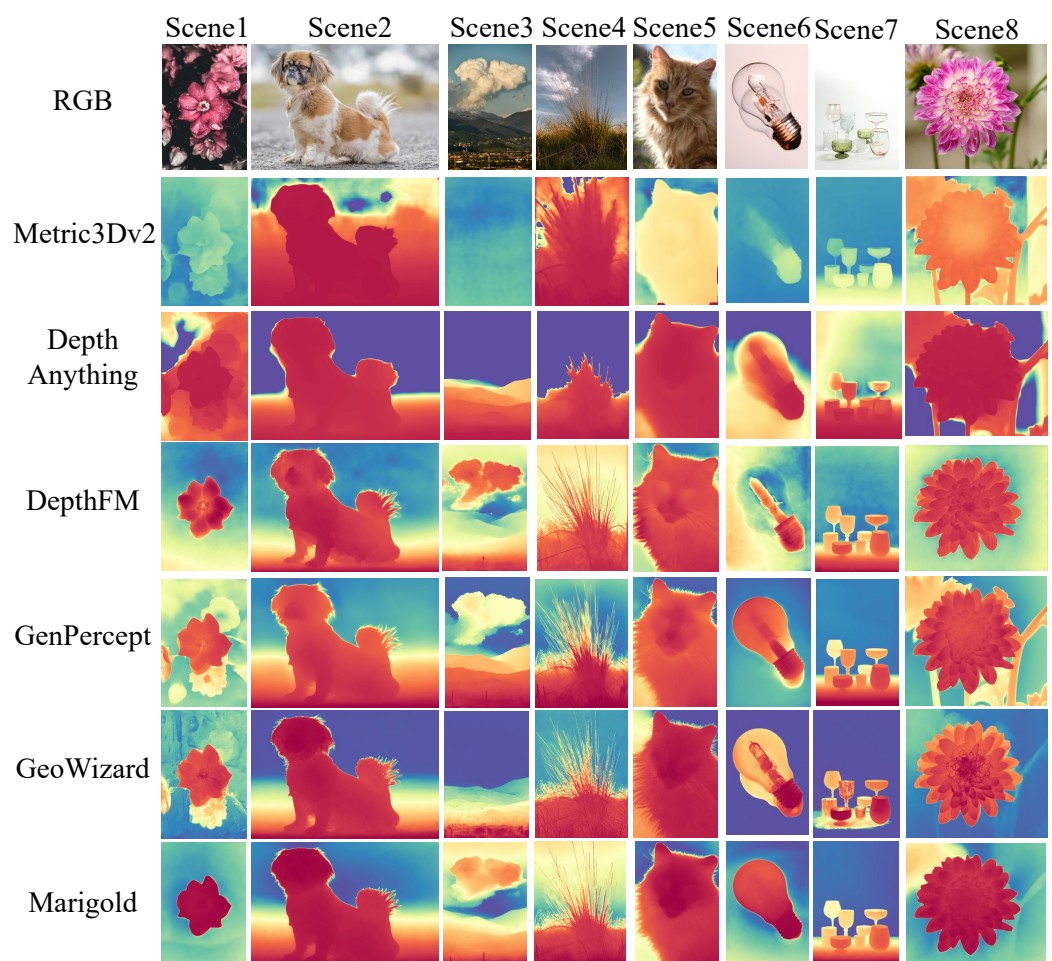

Figure 2: Visualization of different depth estimation methods.

ScanNet Dai et al. (2017) is a real RGB-D video dataset of indoor scenes. We use the ground-truth surface normal and test split (800 sampled images) provided by FrameNet Huang et al. (2019b). To mitigate the noise, it first computes two (X and Y) tangent principal directions by adopting the 4-RoSY field using QuadriFlow Huang et al. (2018) as proposed by TextureNet Huang et al. (2019a), and the ground-truth normal can be directly computed as the cross product of them.

DIODE Vasiljevic et al. (2019) 1024×768 collects both outdoor and indoor scenes. It collects high-quality data, but it contains very low diversity with only 2 scenes for evaluation.

Sintel Butler et al. (2012) is a synthetic dataset derived from an open-source 3D animated short film. We calculate the ground-truth surface normal with the provided ground-truth depth maps and intrinsic parameters following the depth-to-normal procedure of DSINE Bae & Davison (2024).

BEDLAM Black et al. (2023) contains synthetic monocular RGB videos with ground-truth 3D bodies with varying numbers of people in realistic scenes with varied lighting and camera motions. We calculate the ground-truth surface normal with the provided ground-truth depth maps and intrinsic parameters following the depth-to-normal procedure of DSINE Bae & Davison (2024).

Infinigen Raistrick et al. (2023) generates diverse high-quality 3D synthetic scene data, which offers broad coverage of objects and scenes in the natural world with natural phenomena. The surface normal is rendered based on Blender.

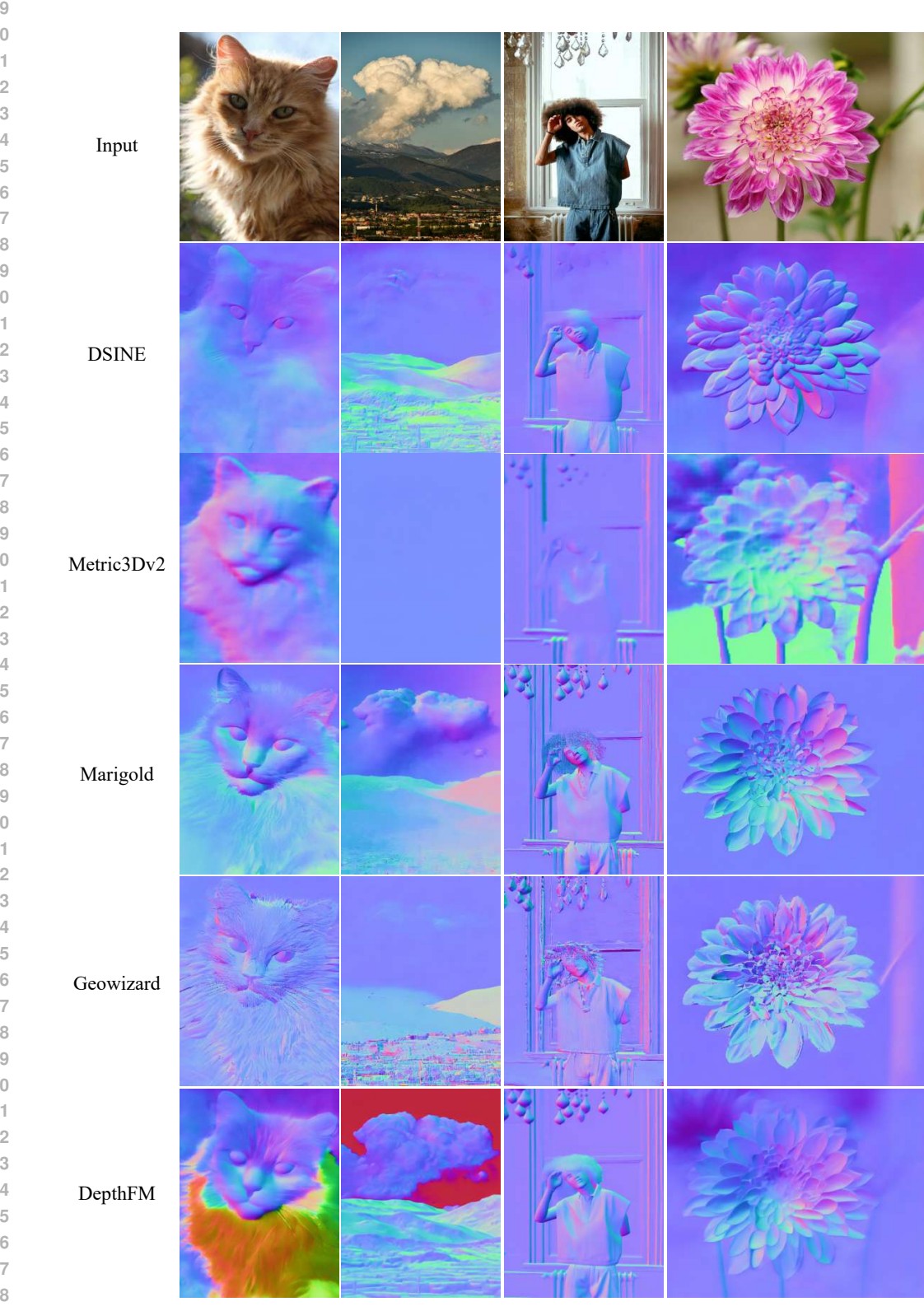

Figure 3: Visualization of different surface normal estimation methods.

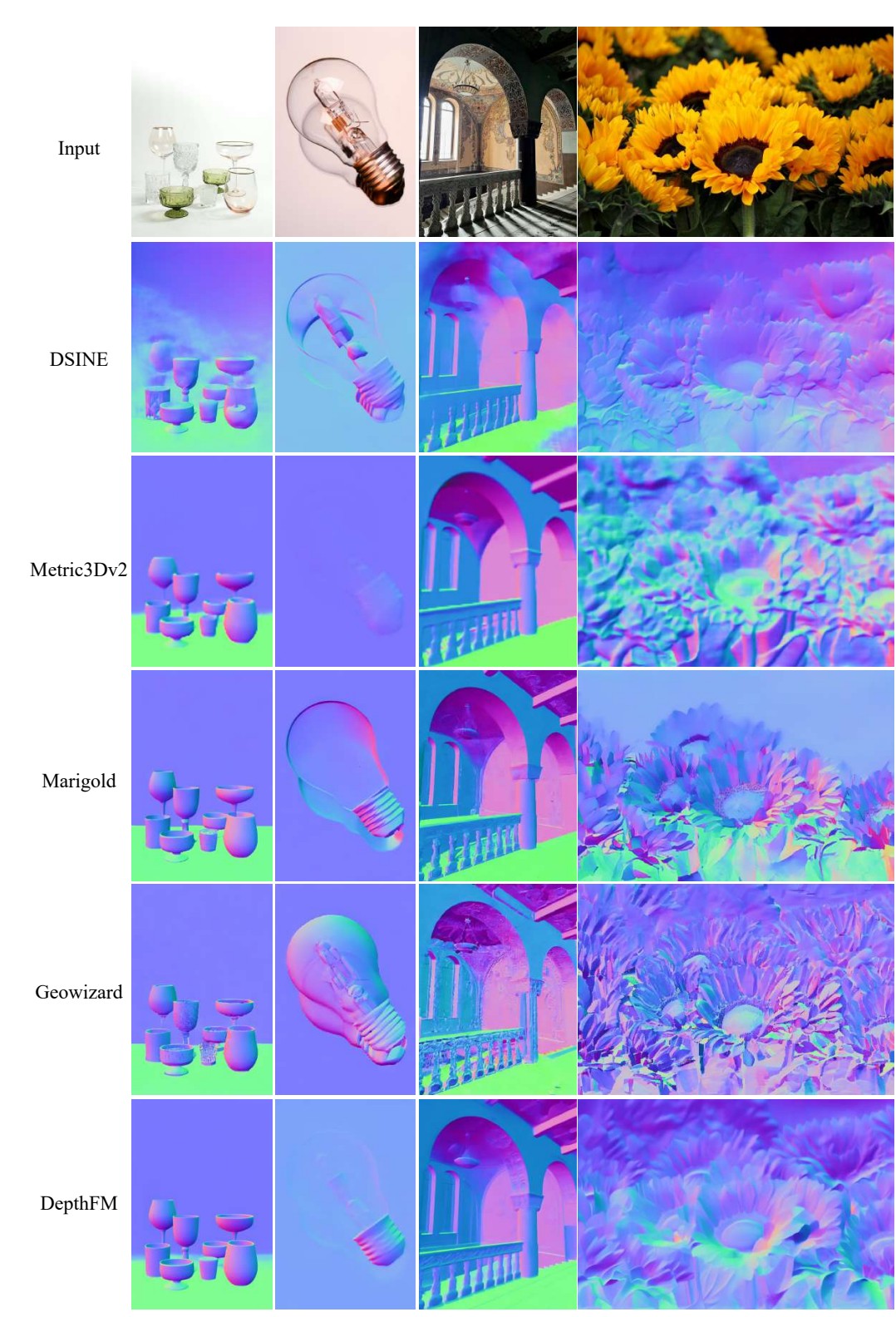

Figure 4: Visualization of different surface normal estimation methods.

Table 1: **Correspondence Estimation Results.** The results are presented for features extracted at different layers with performance binned based on the viewpoint variation for the image pair. 'DA' indicates Depth-Anything. 'DA (77K)' indicates Depth-Anything trained with only 77K synthetic data. 'SD10' indicates Stable Diffusion model inference 10 steps. 'MIX' indicates using a mixture of datasets during the training. The higher the recall in the table, the better the performance.

| Model | Architecture | Dataset | Layers | Spair-71k | | | | Paired ScanNet | | | | NAVI | | | |
|---|---|---|---|---|---|---|---|---|---|---|---|---|---|---|---|
| | | | | $d{=}0$ | $d{=}1$ | $d{=}2$ | all | $\theta_0^{15}$ | $\theta_{15}^{30}$ | $\theta_{30}^{60}$ | $\theta_{60}^{180}$ | $\theta_0^{30}$ | $\theta_{30}^{60}$ | $\theta_{60}^{90}$ | $\theta_{90}^{120}$ |
| *Pre-train Models* | | | | | | | | | | | | | | | |
| DINOv2 | ViT-L14 | LVD | Block0 | 8.5 | 6.2 | 5.3 | 7.5 | 17.2 | 14.1 | 10.1 | 4.7 | 66.2 | 37.2 | 19.6 | 11.5 |
| DINOv2 | ViT-L14 | LVD | Block1 | 25.0 | 14.0 | 10.8 | 19.3 | 29.0 | 20.8 | 13.5 | 5.2 | 92.1 | 57.9 | 25.6 | 12.8 |
| DINOv2 | ViT-L14 | LVD | Block2 | 53.9 | 34.6 | 31.6 | 44.5 | 35.2 | 24.1 | 16.3 | 6.6 | 95.3 | 70.0 | 35.4 | 18.5 |
| DINOv2 | ViT-L14 | LVD | Block3 | 62.8 | 53.3 | 54.2 | 57.2 | 36.5 | 27.0 | 20.8 | 12.2 | 92.2 | 72.3 | 48.9 | 35.0 |
| DINOv2 | ViT-L14+reg | LVD | Block0 | 12.2 | 8.8 | 8.1 | 10.4 | 14.0 | 14.2 | 11.4 | 5.0 | 79.9 | 40.8 | 24.5 | 13.6 |
| DINOv2 | ViT-L14+reg | LVD | Block1 | 41.2 | 22.8 | 17.1 | 32.0 | 52.0 | 39.4 | 23.7 | 9.1 | 95.4 | 65.6 | 32.7 | 15.8 |
| DINOv2 | ViT-L14+reg | LVD | Block2 | 64.2 | 45.9 | 42.4 | 55.0 | 50.6 | 39.3 | 26.2 | 12.0 | 95.2 | 75.0 | 49.1 | 28.6 |
| DINOv2 | ViT-L14+reg | LVD | Block3 | 59.3 | 53.2 | 54.9 | 55.0 | 45.0 | 35.4 | 26.1 | 15.4 | 88.6 | 71.2 | 54.3 | 36.1 |
| SAM | ViT-L16 | SA-1B | Block0 | 9.9 | 6.1 | 5.4 | 8.0 | 14.5 | 9.9 | 7.5 | 3.5 | 78.0 | 43.3 | 20.4 | 11.4 |
| SAM | ViT-L16 | SA-1B | Block1 | 22.6 | 15.8 | 12.5 | 18.3 | 37.2 | 29.7 | 19.7 | 6.2 | 86.4 | 52.0 | 23.8 | 12.5 |
| SAM | ViT-L16 | SA-1B | Block2 | 34.8 | 23.1 | 17.0 | 28.2 | 47.6 | 40.4 | 27.3 | 8.7 | 91.2 | 60.1 | 28.2 | 14.2 |
| SAM | ViT-L16 | SA-1B | Block3 | 30.2 | 18.1 | 13.0 | 24.1 | 52.6 | 43.9 | 28.7 | 9.6 | 88.5 | 57.6 | 26.9 | 13.5 |
| SD10 | UNet | LAION | Block0 | 13.2 | 5.3 | 3.5 | 9.2 | 10.8 | 5.4 | 3.2 | 1.3 | 75.1 | 32.5 | 16.6 | 7.4 |
| SD10 | UNet | LAION | Block1 | 58.6 | 36.4 | 28.6 | 47.8 | 67.0 | 56.1 | 32.0 | 8.7 | 93.4 | 59.7 | 26.2 | 11.4 |
| SD10 | UNet | LAION | Block2 | 24.0 | 16.8 | 13.4 | 20.2 | 61.4 | 49.5 | 28.4 | 9.4 | 79.0 | 42.5 | 22.3 | 12.2 |
| SD10 | UNet | LAION | Block3 | 4.6 | 4.3 | 4.4 | 4.3 | 17.2 | 12.8 | 8.9 | 5.0 | 35.3 | 22.9 | 15.2 | 11.0 |
| *Deterministic Geometry Foundation Models* | | | | | | | | | | | | | | | |
| MiDaS | ViT-L16 | MIX 6 | Block0 | 15.6 | 10.2 | 8.7 | 13.0 | 50.3 | 39.0 | 24.4 | 11.2 | 79.0 | 49.1 | 25.0 | 14.5 |
| MiDaS | ViT-L16 | MIX 6 | Block1 | 27.3 | 22.8 | 23.2 | 24.5 | 56.4 | 47.4 | 31.6 | 13.9 | 83.2 | 56.0 | 32.1 | 21.6 |
| MiDaS | ViT-L16 | MIX 6 | Block2 | 28.2 | 23.4 | 25.1 | 25.5 | 55.5 | 46.0 | 30.8 | 14.3 | 82.2 | 56.3 | 33.1 | 22.9 |
| MiDaS | ViT-L16 | MIX 6 | Block3 | 25.8 | 21.3 | 23.6 | 23.4 | 52.4 | 42.1 | 27.6 | 13.1 | 79.6 | 53.0 | 31.4 | 21.6 |
| DA | ViT-L16 | MIX | Block0 | 8.0 | 6.1 | 5.3 | 6.8 | 21.4 | 17.5 | 12.2 | 5.4 | 66.1 | 35.6 | 20.6 | 12.5 |
| DA | ViT-L16 | MIX | Block1 | 24.4 | 13.8 | 11.1 | 19.4 | 34.2 | 26.4 | 17.0 | 6.1 | 92.4 | 55.9 | 27.7 | 14.0 |
| DA | ViT-L16 | MIX | Block2 | 51.4 | 31.6 | 28.4 | 42.2 | 30.2 | 23.5 | 16.2 | 6.8 | 95.2 | 68.1 | 35.1 | 17.5 |
| DA | ViT-L16 | MIX | Block3 | 58.9 | 48.6 | 49.7 | 53.5 | 29.8 | 21.4 | 16.8 | 9.3 | 90.9 | 67.8 | 47.9 | 30.5 |
| DA(77K) | ViT-L16 | MIX | Block0 | 8.0 | 5.8 | 5.2 | 6.7 | 18.3 | 15.1 | 10.7 | 5.0 | 63.6 | 34.9 | 20.4 | 12.4 |
| DA(77K) | ViT-L16 | MIX | Block1 | 24.2 | 13.6 | 11.0 | 19.1 | 34.4 | 25.7 | 16.6 | 6.4 | 92.4 | 54.6 | 26.9 | 13.7 |
| DA(77K) | ViT-L16 | MIX | Block2 | 50.8 | 31.0 | 28.0 | 41.6 | 43.4 | 32.9 | 23.2 | 8.9 | 94.9 | 67.0 | 34.9 | 17.8 |
| DA(77K) | ViT-L16 | MIX | Block3 | 53.6 | 42.2 | 43.4 | 47.7 | 38.4 | 29.8 | 21.7 | 11.8 | 92.8 | 71.9 | 50.7 | 31.0 |
| Metric3Dv2 | ViT-L16 | MIX | Block0 | 12.0 | 8.6 | 7.9 | 10.1 | 10.2 | 10.5 | 8.7 | 4.3 | 79.1 | 39.7 | 23.5 | 12.7 |
| Metric3Dv2 | ViT-L16 | MIX | Block1 | 39.0 | 22.0 | 16.0 | 30.7 | 55.7 | 42.8 | 25.2 | 8.8 | 94.2 | 61.4 | 29.6 | 14.2 |
| Metric3Dv2 | ViT-L16 | MIX | Block2 | 60.2 | 41.5 | 39.8 | 51.6 | 63.1 | 54.7 | 36.8 | 14.8 | 94.1 | 68.1 | 36.9 | 20.9 |
| Metric3Dv2 | ViT-L16 | MIX | Block3 | 53.6 | 42.3 | 42.8 | 48.0 | 59.5 | 50.3 | 35.1 | 16.3 | 86.6 | 56.5 | 29.6 | 17.2 |
| *Generative Geometry Foundation Models* | | | | | | | | | | | | | | | |
| Marigold | UNet | MIX | Block0 | 14.0 | 4.6 | 3.5 | 9.6 | 8.4 | 5.8 | 3.4 | 1.3 | 81.7 | 37.0 | 17.0 | 8.0 |
| Marigold | UNet | MIX | Block1 | 53.8 | 29.5 | 23.7 | 42.5 | 42.2 | 32.4 | 18.7 | 4.4 | 92.8 | 59.1 | 25.6 | 11.7 |
| Marigold | UNet | MIX | Block2 | 27.2 | 15.8 | 12.5 | 21.3 | 45.5 | 34.1 | 18.1 | 5.4 | 83.5 | 45.4 | 21.5 | 11.2 |
| Marigold | UNet | MIX | Block3 | 8.0 | 6.4 | 6.3 | 7.1 | 18.0 | 12.8 | 7.5 | 3.5 | 43.3 | 25.2 | 15.9 | 9.8 |
| DepthFM | UNet | MIX | Block0 | 20.0 | 8.4 | 6.1 | 14.3 | 23.1 | 16.4 | 7.4 | 2.2 | 85.9 | 40.6 | 17.0 | 8.0 |
| DepthFM | UNet | MIX | Block1 | 50.8 | 31.4 | 25.2 | 42.1 | 46.4 | 39.1 | 24.0 | 7.2 | 94.1 | 62.4 | 29.2 | 13.0 |
| DepthFM | UNet | MIX | Block2 | 22.6 | 13.8 | 10.7 | 18.8 | 46.0 | 36.7 | 20.0 | 6.2 | 80.5 | 41.7 | 20.7 | 11.0 |
| DepthFM | UNet | MIX | Block3 | 3.9 | 3.5 | 3.0 | 3.6 | 11.2 | 8.4 | 6.3 | 3.8 | 39.0 | 25.6 | 16.3 | 10.4 |
| GeowizardD | UNet | MIX | Block0 | 13.7 | 4.7 | 2.9 | 9.7 | 8.0 | 5.1 | 3.2 | 1.34 | 81.9 | 35.1 | 16.6 | 8.5 |
| GeowizardD | UNet | MIX | Block1 | 41.3 | 19.1 | 13.4 | 31.2 | 43.0 | 32.5 | 16.9 | 3.8 | 89.3 | 52.3 | 22.5 | 10.7 |
| GeowizardD | UNet | MIX | Block2 | 20.2 | 11.4 | 8.4 | 16.3 | 38.3 | 27.1 | 12.8 | 3.7 | 71.1 | 35.4 | 17.8 | 10.1 |
| GeowizardD | UNet | MIX | Block3 | 8.5 | 5.7 | 5.8 | 7.2 | 13.8 | 9.9 | 5.4 | 2.7 | 32.5 | 20.1 | 12.7 | 8.1 |
| GeowizardN | UNet | MIX | Block0 | 11.1 | 3.6 | 2.9 | 7.6 | 8.5 | 5.1 | 3.0 | 1.3 | 80.6 | 33.9 | 15.1 | 7.6 |
| GeowizardN | UNet | MIX | Block1 | 43.3 | 20.2 | 15.5 | 32.8 | 48.6 | 37.8 | 20.8 | 5.0 | 88.8 | 53.7 | 22.4 | 10.4 |
| GeowizardN | UNet | MIX | Block2 | 22.5 | 12.3 | 9.4 | 18.0 | 43.4 | 32.8 | 16.3 | 4.5 | 68.9 | 36.6 | 17.5 | 9.2 |
| GeowizardN | UNet | MIX | Block3 | 6.8 | 5.4 | 4.8 | 6.2 | 13.0 | 10.2 | 6.3 | 2.7 | 27.5 | 17.6 | 12.0 | 7.3 |
| GenPercept | UNet | MIX | Block0 | 21.5 | 9.6 | 7.0 | 16.0 | 22.7 | 16.1 | 7.1 | 1.8 | 84.4 | 40.8 | 17.3 | 8.0 |
| GenPercept | UNet | MIX | Block1 | 62.0 | 41.9 | 34.4 | 52.2 | 55.7 | 46.4 | 27.8 | 6.4 | 94.5 | 64.9 | 29.7 | 13.3 |
| GenPercept | UNet | MIX | Block2 | 28.2 | 16.4 | 13.3 | 22.9 | 54.9 | 43.0 | 23.8 | 6.1 | 84.5 | 45.1 | 21.5 | 10.8 |
| GenPercept | UNet | MIX | Block3 | 8.0 | 5.9 | 5.9 | 7.0 | 26.6 | 19.0 | 10.6 | 3.8 | 58.3 | 31.4 | 17.4 | 10.0 |

MuSHRoom Ren et al. (2024) is an indoor real-world multi-sensor hybrid room dataset, which contains 10 rooms captured by Kinect, iPhone, and Faro scanner. We use the ground-truth normal annotations supported by gaustudio Ye et al. (2024).

Tank and Temples (T&T) Knapitsch et al. (2017) is a dataset including both outdoor scenes and indoor environments, whose ground-truth data is captured using an industrial laser scanner. We use the ground-truth normal annotations supported by gaustudio Ye et al. (2024).

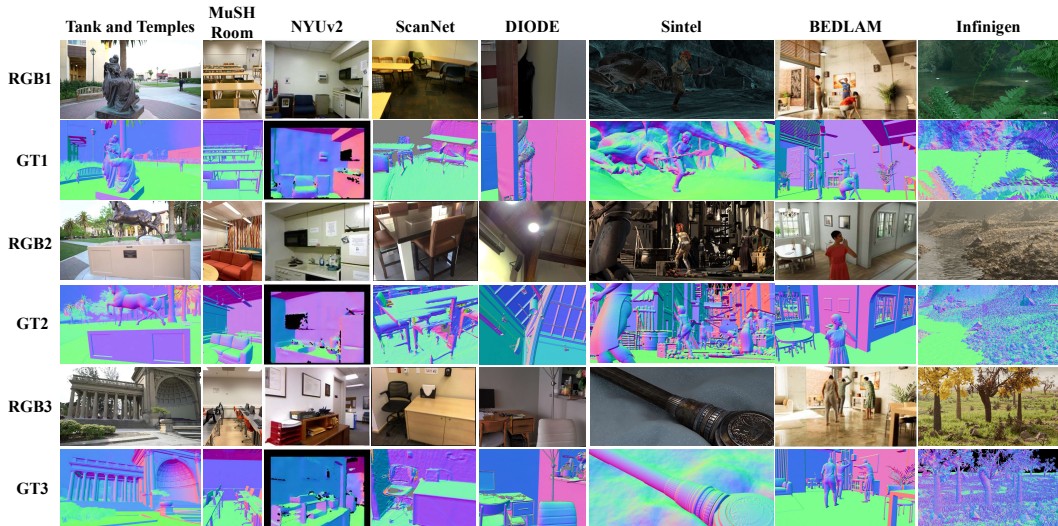

Figure 5: Visualization of the ground-truth surface normal.

## A.5    POINT CLOUD VISUALIZATION

In this section, we visualize affine-invariant depth estimation results of Marigold, DepthAnything, and our fine-tuned DINOv2 with DPT head model on NuScenes and Waymo datasets. Concretely, we calculate the scale and shift values with the ground truth in the dataset, then we reproject the depth map into the 3D point cloud format. The visualization again demonstrates that the models fine-tuned with small-scale synthetic data, *i.e.*, Marigold and DINOv2 with DPT head, are comparable with Depth Anything in the wild scenes.

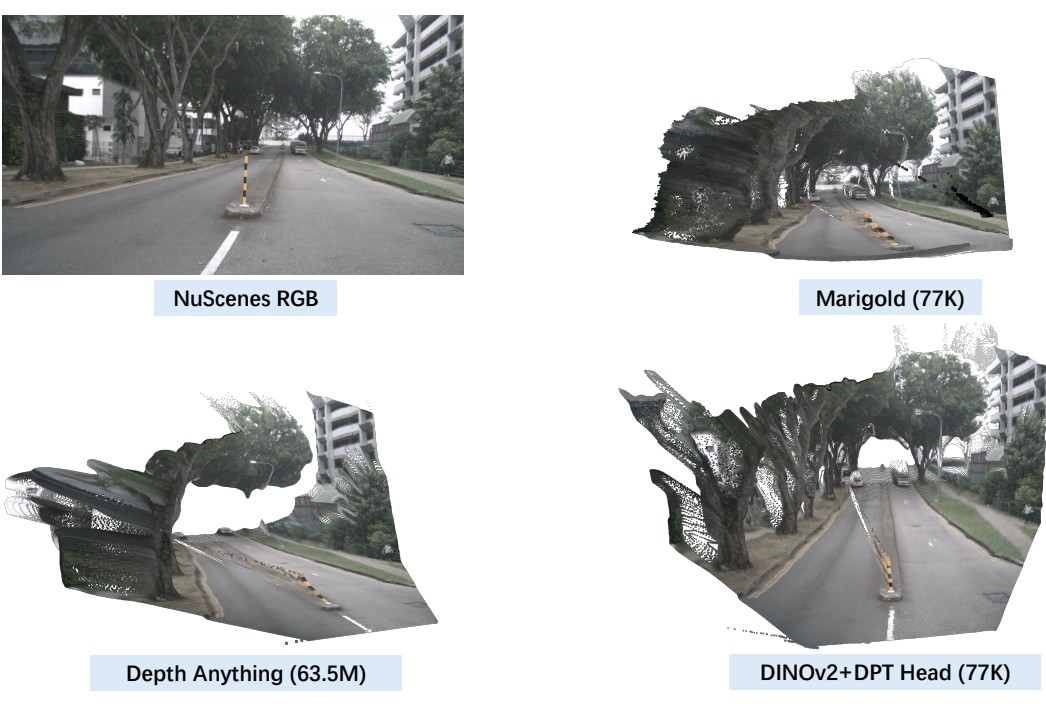

Figure 6: Point cloud visualization on NuScenes Dataset.

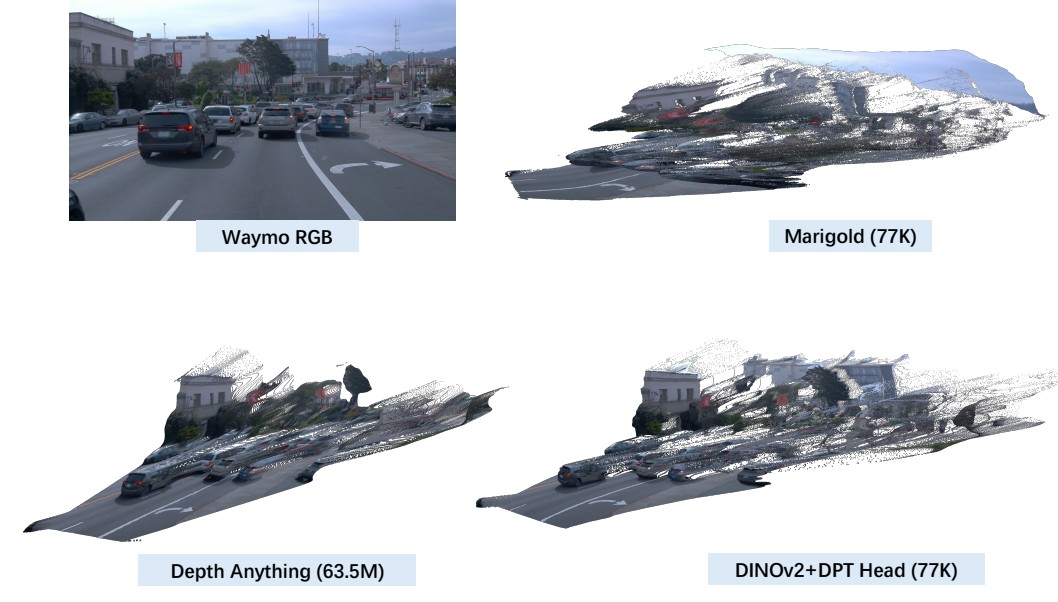

Figure 7: Point cloud visualization on Waymo Dataset.

## A.6 LIMITATIONS AND FUTURE WORKS

The discussion of monocular depth estimation in this work is limited to single-image monocular affine-invariant depth estimation and monocular metric depth estimation. Video-based depth estimation is also an important topic, we leave it for future exploration.

## A.7 BROADER IMPACTS

In this section, we aim to discuss the potential societal impacts. The positive societal impacts encompass two aspects. First, it helps the research community gain in-depth knowledge about monocular geometry estimation, including performance comparisons between different models, technical details of current models, and future approaches. The release of this work also helps researchers perform experiments to evaluate their methods more comprehensively, fairly, and conveniently. Furthermore, it will significantly boost the progress of downstream tasks. As we mentioned in the paper, monocular geometry estimation can be applied to many downstream tasks, thereby accelerating their progress. In summary, we believe this work will have substantial positive effects on the research community, enriching the capacity of current and future applications and products, and ultimately improving people's lives. We also evaluated the negative societal impacts and found none.