# OpenReview forum: "Benchmarking and Analyzing Monocular Geometry Estimation Models"
_ICLR.cc/2025/Conference — ICLR 2025 Conference Withdrawn Submission_

### Official Review · Reviewer_4cdN · 2024-10-22

**Soundness:** 2
**Presentation:** 1
**Contribution:** 1
**Rating:** 3
**Confidence:** 4

**Summary:**

The paper is a survey on some recent monocular depth estimation and surface normal estimation methods, in particular such models that are derived via finetuning from DINOv2 or Stable Diffusion 2.1 foundation models. The paper runs several comparison experiments between these methods with different fine-tuning modes and tries to derive some conclusive insights from these experiments.

**Strengths:**

I generally like papers that run a deeper analysis on existing methodology. This paper aims a bit in this direction, unfortunately not in a sufficiently rigorous way (see weaknesses). One small positive step in this direction I noticed in conjunction with the decoder supervision when training the surface normal estimation and the intervention by modifying the GenPercept method.

**Weaknesses:**

1. The paper lacks clarity and is hard to read. I was wondering actively why after each sentence I read I had problems remembering what I just read. The result of my analysis: the sentences use strange or vague terminology with insufficient context and were not to the point. The same is true for whole paragraphs. Information is spread all over the place without any help given to the reader to make sense of it. This makes it very tedious to read this paper. It was no fun reading it. I did it because I had to.

2. There are only vague formulations when it comes to the contributions. For example, it is claimed that "we build fair and strong baselines", but when reading on, the baselines are simply existing models that have been finetuned on a different data subset. This indeed allows a more direct comparison, but nothing new is built here. Another example on benchmarks: "we build more diverse scenes with high-quality labels for geometry evaluation. For depth estimation, we introduce three extra benchmark datasets", but in fact these datasets already exist and were simply used for the experiments. Again, nothing new is built here.

3. It would be okay with nothing new being built, if this paper had a strong scientific side. There was some hope from the abstract, which talks about many insights. However, the insights are anecdotal and only mildly conclusive. Except one attribution to decoder supervision, there is no analysis of the causes of a certain behavior. Therefore, the findings can be due to multiple reasons not controlled in the experiments. As a result, there is nothing conclusive I can take home. The conclusion by the authors "We identify that a strong pre-train
model, either Stable Diffusion or DINOv2, combined with high-quality fine-tuning data, is the key to achieving generalizable monocular geometry estimation." doesn't really tell anything. Only Stable Diffusion and DINOv2 were tested, so it is impossible to identify that they are really important, and that high-quality fine-tuning data (what does high-quality mean exactly?) is important for good results is not a new finding.

**Questions:**

How can I see from Table 7 that "high-quality fine-tuning data" was used or not used? What makes fine-tuning data "high-quality"?

What is, from your point of view, the strongest finding in this paper? Why?

---

### Official Review · Reviewer_zbWf · 2024-11-02

**Soundness:** 3
**Presentation:** 3
**Contribution:** 2
**Rating:** 5
**Confidence:** 4

**Summary:**

This paper examines factors affecting geometry estimation model performance, comparing discriminative and generative pretraining methods. While discriminative models need extensive fine-tuning data, generative methods can generalize using minimal synthetic data. To fairly assess models, the authors establish unified baselines. Results suggest that simpler one-step fine-tuning suffices for generative tasks like depth and surface normal estimation, with DINOv2 discriminative models slightly outperforming generative ones. Additionally, metric depth estimation requires more fine-tuning data than scale-invariant tasks. This study aims to guide improvements in geometry estimation models.

**Strengths:**

The article is well-written, clear, and easy to understand. It presents extensive comparative experiments to analyze and contrast various methods thoroughly. The paper offers several practical and insightful conclusions that contribute to advancing the field. For example, the findings suggest that the diffusion-based protocol may not be optimal for fine-tuning generative models. Instead, a simpler one-step fine-tuning approach is sufficient for tasks such as depth and surface normal estimation. Notably, both discriminative and generative pretraining methods perform effectively with small-scale, high-quality data, though discriminative models like DINOv2 show a slight advantage over generative counterparts. Additionally, the study highlights that metric depth estimation requires significantly more fine-tuning data than scale-invariant depth estimation to accurately capture depth scale. These findings provide valuable guidance for future developments in geometry estimation research.

**Weaknesses:**

1. The article resembles an experimental report more than a formal academic paper. While it thoroughly documents the comparisons, it lacks the depth and structure typically expected in scholarly research.

2. Lack of novelty. Although the paper offers valuable recommendations, it falls short of proposing specific solutions or implementing them, which limits its contribution to advancing methodology.

3. While the paper appears to aim at establishing a new benchmark, it does not introduce a new dataset but rather relies on existing datasets. This reliance on pre-existing data weakens the contribution toward setting a truly novel benchmark, as it doesn’t address potential limitations or gaps within current datasets that new benchmarks typically seek to fill.

**Questions:**

Can you explain this expression in Line 64 to 65: The performance distinction between discriminative and generative geometry estimation models when trained on the same scale and quality of data also remains unclear.

---

### Official Review · Reviewer_8cfc · 2024-11-03

**Soundness:** 3
**Presentation:** 4
**Contribution:** 4
**Rating:** 6
**Confidence:** 4

**Summary:**

This paper thoroughly experiments with and compares existing state-of-the-art (SOTA) monocular depth estimation methods, with the experimental comparisons covering both discriminative and generative pretraining methods. The paper not only categorizes and compares the methods in terms of technical details but also contrasts them using the released weights of each method. Additionally, the paper re-trains the models under consistent experimental conditions to compare the performance differences among the methods. It provides a comprehensive summary and comparison of the existing methods. Furthermore, the paper presents some effective conclusions that serve as guidance for subsequent monocular depth estimation research and the authors build a unified code base, which may promote community development.

**Strengths:**

1. Exhaustive experiments have led to a wealth of meaningful conclusions
2. Comparisons based on the same training conditions are helpful in identifying superior structures, loss functions, and training methods.

**Weaknesses:**

W1:  L419 'However, the performance of small-scale dataset finetuning is largely behind the official model trained on 16M data samples. Hence, large-scale datasets with diverse scales and cameras is still indispensable for metric depth estimation.' This conclusion is not convincing enough, since the original Metric3D V2 has been trained for 800k iterations with a batch of 192, while for the retrained model, the training iteration is 30k with unknown batch number (maybe 96 mentioned in L397?).

W2: About the evaluation metrics. Since the authors pay attention to the current zero-shot depth estimation and normal estimation methods, which pay much attention to 'robustness', maybe a metric about the error variance is needed to represent the robustness of each model on the benchmark. For example, now the AbsRel and delta_1 are all the mean values, maybe you can present the variance of AbsRel and delta_1 to present the model robustness.

W3: the overall performance is needed. Since most of the Tables are evaluated through several datasets/benchmarks, a overall result in each table is needed to present the model performance.

W4: most of the experiments are conducted under insufficient training (20k iterations), which may affect the validity of the experimental conclusions.

**Questions:**

Q1: L404: ‘We can see from Table 7 that our discriminative model trained on 77K data outperforms Metric3Dv2 (Hu et al., 2024) in all three
datasets’. How to get that conclusion? According to Table7, it seems that 'ViT+DPT Head' model performs poorly on MatrixCity dataset, with its performance significantly lower than that of Metric3D v2.

Q2: In Sec. 4.1 and Sec. 4.2 texts, are the Table 9 and Table 10 incorrectly referenced?

Q3: Why some methods in Table 9 are not included in Table 10?

---

### Official Review · Reviewer_yb7J · 2024-11-04

**Soundness:** 3
**Presentation:** 2
**Contribution:** 2
**Rating:** 5
**Confidence:** 4

**Summary:**

The paper integrates the latest depth estimation models, dividing them into discriminative-based and generative-based approaches, and evaluates them using diverse datasets, including new benchmark datasets created by the authors. It conducts extensive experiments on various tasks, such as general depth estimation, metric depth, surface normal estimation, and cross-view consistency, analyzing the strengths and weaknesses of each model. Additionally, the paper explores different fine-tuning methods and experiments with lighter network architectures, providing practical insights for the next depth estimation model development.

**Strengths:**

- Expensive experiments: The paper conducts a thorough comparison of discriminative and generative geometry models using a wide range of datasets and models, evaluating their performance under different training conditions. This comprehensive approach provides valuable assistance to researchers, allowing them to follow the latest depth estimation models and gain insights from the detailed comparisons.

- Practical Insights: The paper highlights the strengths and weaknesses of specific models and approaches, providing practical implications for future research. The observation that Marigold demonstrates greater robustness and good generalization capabilities on new data is interesting.

**Weaknesses:**

- Lack of Depth in Analysis: The paper often bases its conclusions on somewhat limited experimental results. For example, it concludes that complex fine-tuning protocols are not necessary based solely on the strong performance of GenPercept's one-step fine-tuning. However, this could be a risky generalization. This conclusion might easily be challenged if a more effective, complex fine-tuning method is proposed in the future. Rather than relying solely on performance metrics, a more in-depth analysis of GenPercept’s network architecture and training dynamics is needed, along with validation under diverse experimental conditions. For instance, evaluating the robustness of different types of datasets or under significant viewpoint changes could enhance the reliability of this conclusion.

- Novelty: Although the paper provides important insights through extensive experimentation, these insights do not introduce entirely new directions. The study effectively aggregates and compares recent research to showcase the current leading approaches. However, this approach leans more towards a re-evaluation of existing work rather than proposing novel methodologies or technological breakthroughs. Therefore, it is questionable whether such comparisons alone are sufficient as substantial contributions.

**Questions:**

- Do you think testing 3D consistency solely with the method proposed by Probe3D is sufficient?
- How can we design a depth estimation model that improves 3D consistency?
- Are you suggesting that overall, data quality is more important than quantity, but for metric depth estimation, both data quantity and quality are essential?
- What kind of networks, designs, and datasets would be good to create and train when constructing new models for relative depth estimation, metric depth estimation, surface normal estimation, and 3D consistency? The summary of suggested insight will be beneficial.

---

### Note · Authors · 2024-11-15

**Comment:**

Dear Reviewers,

Thank you for your feedback on our submission. However, it appears that the full value and potential impact of our work have not been fully appreciated, particularly by Reviewer 4cdN. Our paper offers significant contributions in advancing the understanding of datasets, pretraining strategies, and model architectures, and we believe it provides valuable insights that are highly relevant to the field.

Given this, we have decided to withdraw our submission and will seek another conference that better aligns with the significance of our work.

Best regards

**Withdrawal Confirmation:**

I have read and agree with the venue's withdrawal policy on behalf of myself and my co-authors.